# Specification of neural circuit architecture shaped by context-dependent patterned LAR-RPTP microexons

Kyung Ah Han [1,2,6], Taek-Han Yoon[1,6], Jinhu Kim [1], Jusung Lee[3], Ju Yeon Lee [4], Gyubin Jang [1,2], Ji Won Um [1,2], Jong Kyoung Kim [3,5] & Jaewon Ko [1,2] ✉

LAR-RPTPs are evolutionarily conserved presynaptic cell-adhesion molecules that orchestrate multifarious synaptic adhesion pathways. Extensive alternative splicing of LAR-RPTP mRNAs may produce innumerable LAR-RPTP isoforms that act as regulatory "codes" for determining the identity and strength of specific synapse signaling. However, no direct evidence for this hypothesis exists. Here, using targeted RNA sequencing, we detected LAR-RPTP mRNAs in diverse cell types across adult male mouse brain areas. We found pronounced cell-type–specific patterns of two microexons, meA and meB, in *Ptprd* mRNAs. Moreover, diverse neural circuits targeting the same neuronal populations were dictated by the expression of different *Ptprd* variants with distinct inclusion patterns of microexons. Furthermore, conditional ablation of *Ptprd* meA[+] variants at presynaptic loci of distinct hippocampal circuits impaired distinct modes of synaptic transmission and object-location memory. Activity-triggered alterations of the presynaptic *Ptprd* meA code in subicular neurons mediates NMDA receptor-mediated postsynaptic responses in CA1 neurons and object-location memory. Our data provide the evidence of cell-type- and/or circuit-specific expression patterns in vivo and physiological functions of LAR-RPTP microexons that are dynamically regulated.

Alternative splicing of precursor mRNAs, an often evolutionarily conserved process by which cells expand molecular repertoires and complexity of the eukaryotic proteome, is remarkably prevalent in the central nervous system (CNS)[1–5]. Recent studies have contributed to our understanding of the mechanisms and functions of alternative splicing events in shaping cell surface recognition, protein-protein interactions, and diverse aspects of neuronal development[6]. Intriguingly, differential alternative splicing regulation exclusively targets transcripts that encode synaptic proteins and build neuronal architectures[1,4]. Specific synaptic connections arise from discrete steps that mandate combinatorial *trans*-synaptic interactions between pre- and postsynaptic neurons[5,7,8]. Thus, alternative splicing of pleiotropic *trans*-synaptic adhesion molecules likely enables an enormous diversity of neural circuits by producing a large number of protein isoforms as presumptive synaptic adhesion codes[5,9].

Leukocyte common antigen-related receptor protein tyrosine phosphatases (LAR-RPTPs), like neurexins (Nrxns), are evolutionarily conserved and are expressed and function at the presynaptic active

[1]Department of Brain Sciences, Daegu Gyeongbuk Institute of Science and Technology (DGIST), Daegu 42988, Korea. [2]Center for Synapse Diversity and Specificity, DGIST, Daegu 42988, Korea. [3]Department of New Biology, DGIST, Daegu 42988, Korea. [4]Korea Basic Science Institute, Research Center for Bioconvergence Analysis, Cheongju 28119, Korea. [5]Department of Life Sciences, Pohang University of Science and Technology (POSTECH), Pohang 37673, Korea. [6]These authors contributed equally: Kyung Ah Han, Taek-Han Yoon. ✉e-mail: jaewonko@dgist.ac.kr

zone[8,10–13]. Moreover, they bind to multifarious ligands that do not overlap with Nrxn ligands and drive the assembly of molecular machinery responsible for presynaptic differentiation[8,12,14,15]. Three members of the vertebrate LAR-RPTP family—PTPσ, PTPδ, and LAR—exhibit similar domain architectures, notably including multiple splice sites, namely, meA–D[11]. meA (9–27 nucleotides [nt]) and meB (12 nt) sites, located in immunoglobulin domains of LAR-RPTP proteins, have received considerable attention owing to the presence of an insert at the meA and/or meB that controls the ability to bind to specific ligands, analogous to the action of Nrxn SS#4[8]. Unlike Nrxn SS#4, meA and meB are categorized as microexons, representing only ~1% of alternative splicing observed, but considered to perform conserved neuronal-specific functions[16–18]. Remarkably, misregulation of alternative splicing microexons has been reported in individuals with neurodevelopmental disorders, including autism spectrum disorders[19–23], illustrating the importance of proper synaptic adhesion networks in nervous system development. Indeed, one prior study showed that PTPδ meA plays an important role in regulating excitatory synapses and non-REM sleep[24]. However, the diversity of LAR-RPTP mRNAs expressed has not been determined with sufficient clarity to establish the quantitative expression of specific LAR-RPTP isoforms in particular brain regions, neuron types, or sets of neural circuits. Obtaining information on the spatiotemporal dynamics of small-sized microexons has been challenging owing to the lack of reliable RNA sequencing (RNA-seq) methodology and computational platforms[25,26].

Here, we employed targeted RNA-seq of LAR-RPTP mRNAs (*Ptprs*, *Ptprd*, and *Ptprf* [encoding PTPσ, PTPδ, and LAR, respectively]) in conjunction with quantitative polymerase chain reaction (PCR) analyses to discriminate individual LAR-RPTP variants to address the following previously unanswered questions: (1) Does LAR-RPTP diversity manifest across brain regions? (2) What is the predominant LAR-RPTP mRNA species in specific brain regions and specific types of neurons? (3) Are LAR-RPTP transcription profiles shared within neural circuits encompassing identical neuronal populations? (4) Are cellular LAR-RPTP transcription profiles static or remodeled by behavioral experience, as is the case for *Nrxn* SS#4? Strikingly, we found a divergence in LAR-RPTP expression profiles in diverse contexts and established that plasticity of PTPδ meA is prominent and can be altered by exposure to environmental stimuli. We further determined the physiological significance of PTPδ meA in regulating specific modes of excitatory synaptic transmission in a circuit-dependent manner and in mediating proper object-location memory. These results collectively build a model in which LAR-RPTPs utilize complex modes of constitutive and alternative inclusions of microexons to shape combinatorial synaptic adhesion pathways in distinct neural circuits.

## Results

### Targeted RNA-seq analyses of LAR-RPTP mRNA transcripts and expression profiling of LAR-RPTP microexons

Despite numerous prior studies (summarized in Supplementary Table 1), it remains unclear which specific microexon-including or –excluding LAR-RPTP variants are expressed across diverse brain regions, and how their mRNA expression varies quantitatively. To determine the repertoire of microexon inclusion patterns among LAR-RPTP mRNAs, we performed targeted RNA-seq of six adult male mouse brain sub-regions—olfactory bulb, cortex, hippocampus, striatum, thalamus, and cerebellum—validating dissection of each area by quantitative PCR (qPCR) using known marker probes (Fig. 1a, Supplementary Fig. 1a, b). Three replicates were sequenced after target enrichment using probes designed to cover > 98% of targeted mRNA transcripts that include LAR-RPTP mRNAs (Fig. 1b).

To characterize the alternative splicing patterns of targeted mRNAs, we analyzed RNA-seq data. The analysis was validated by performing an initial examination of the splicing pattern of the known exon, SS4, in *Nrxn1* as a positive control. The detectability of the

splicing event associated with the known SS4 in *Nrxn1* was determined by evaluating local splicing variations (LSVs) of *Nrxn1* in six brain regions using Modeling Alternative Junction Inclusion Quantification (MAJIQ)[27]. Two LSVs containing SS4 were detected: a single-source LSV (SS-LSV) and a single-target LSV (ST-LSV). The SS-LSV used exon 21 as a reference exon spliced to downstream exon 22 (SS4), whereas ST-LSV used exon 23 as the reference exon spliced to upstream exon 22 (SS4) (Supplementary Fig. 1c, d). A higher relative inclusion of SS4 in the cerebellum and olfactory bulb compared with the cortex region was observed for both LSVs, a finding consistent with a previous report[28].

Next, we used MAJIQ to investigate the presence of known alternative splicing sites within four microexon segments (meA–meD) in LAR-RPTP mRNAs, as visualized using the *DEXSeq*[29]. First, to evaluate the exon utilization threshold for LAR-RPTP mRNAs, we empirically set the read count cutoff at 10 and performed RT-PCRs and sequencing validation for exons with fewer than 30 read counts (Supplementary Fig. 1e). We found that exons with larger than 10 normalized read counts were reliably detected. This analysis successfully detected short nucleotide sequences encoding meA, meB, and meC peptides in all three mouse LAR-RPTP mRNAs (Fig. 1c, Supplementary Fig. 1f–h). However, short nucleotide sequences encoding meD were not detected in mouse *Ptprs* (Supplementary Fig. 1f), suggesting differences in the alternative splicing mechanism between mice and humans[30]. In addition, eight previously uncharacterized exons in *Ptprd* and one exon in *Ptprf* were identified, whereas no such exons were observed in *Ptprs* (Supplementary Fig. 1f–h). Notably, we made the intriguing discovery of a previously unrecognized microexon located between the last fibronectin type III repeat and transmembrane domain in both *Ptprd* (exon 77) and *Ptprf* (exon 25) (Supplementary Fig. 1g, h). Among the exons captured by targeted RNA-seq, we specifically focused on meA and meB sequences of LAR-RPTP mRNAs owing to their involvement in protein-protein interactions and potential significance in neuronal functions[11,19] (Fig. 1d). To quantitatively analyze the splicing patterns of these microexons in six different brain regions, we used MAJIQ to estimate the percent spliced index (PSI) for *Ptprs* meA, *Ptprs* meB, *Ptprd* meB, *Ptprf* meA, and *Ptprf* meB (Fig. 1e). However, for *Ptprd* meA, which consists of the individual microexons meA1 and meA2, we used kallisto[31] instead of MAJIQ because it better captured the precise microexon splicing patterns (Fig. 1e). To further validate the results obtained with MAJIQ, we conducted additional analyses of the splicing patterns of *Ptprs* meA, *Ptprs* meB, *Ptprd* meB, *Ptprf* meA, and *Ptprf* meB using kallisto (Supplementary Fig. 2). The outcomes were comparable between MAJIQ and kallisto, reinforcing our confidence in the robustness of these analyses.

*Ptprs* meA+ variants were barely expressed in the examined brain regions, whereas detectable expression was noted in the olfactory bulb (Fig. 1e). *Ptprd* meA variants (meA1+meA2+, meA1+meA2−, meA1−meA2+, and meA1−meA2−) and *Ptprf* meA variants showed heterogeneous expression profiles across the six brain regions. *Ptprs* meB+ variants showed high expression in the thalamus and moderate expression in other brain regions. Notably, *Ptprd* mRNAs were predominantly expressed as meB+ isoforms in all brain regions examined, whereas *Ptprf* mRNAs were predominantly expressed as meB− isoforms.

To validate the RNA-seq findings, we designed RT-PCR reactions targeting junction sequences specific to meA or meB (Fig. 1d). Subsequent analyses of the resulting RT-PCR products on polyacrylamide gels to discriminate PCR products containing or excluding small-sized microexons (a method termed as DNA-PAGE), as previously described[32], confirmed the inclusion or exclusion of the microexons, consistent with the RNA-seq data and supporting our observations (Fig. 1e–g). Furthermore, the results of the RT-PCR products were strongly correlated with the expected PSI (E[Ψ]) values calculated from the RNA-seq data, with a Pearson correlation coefficient of 0.9565 (Fig. 1h). Except for the *Ptprd* meA+ and *Ptprf* meA+ variants, the LAR-RPTP mRNA microexon expression patterns showed quite similar

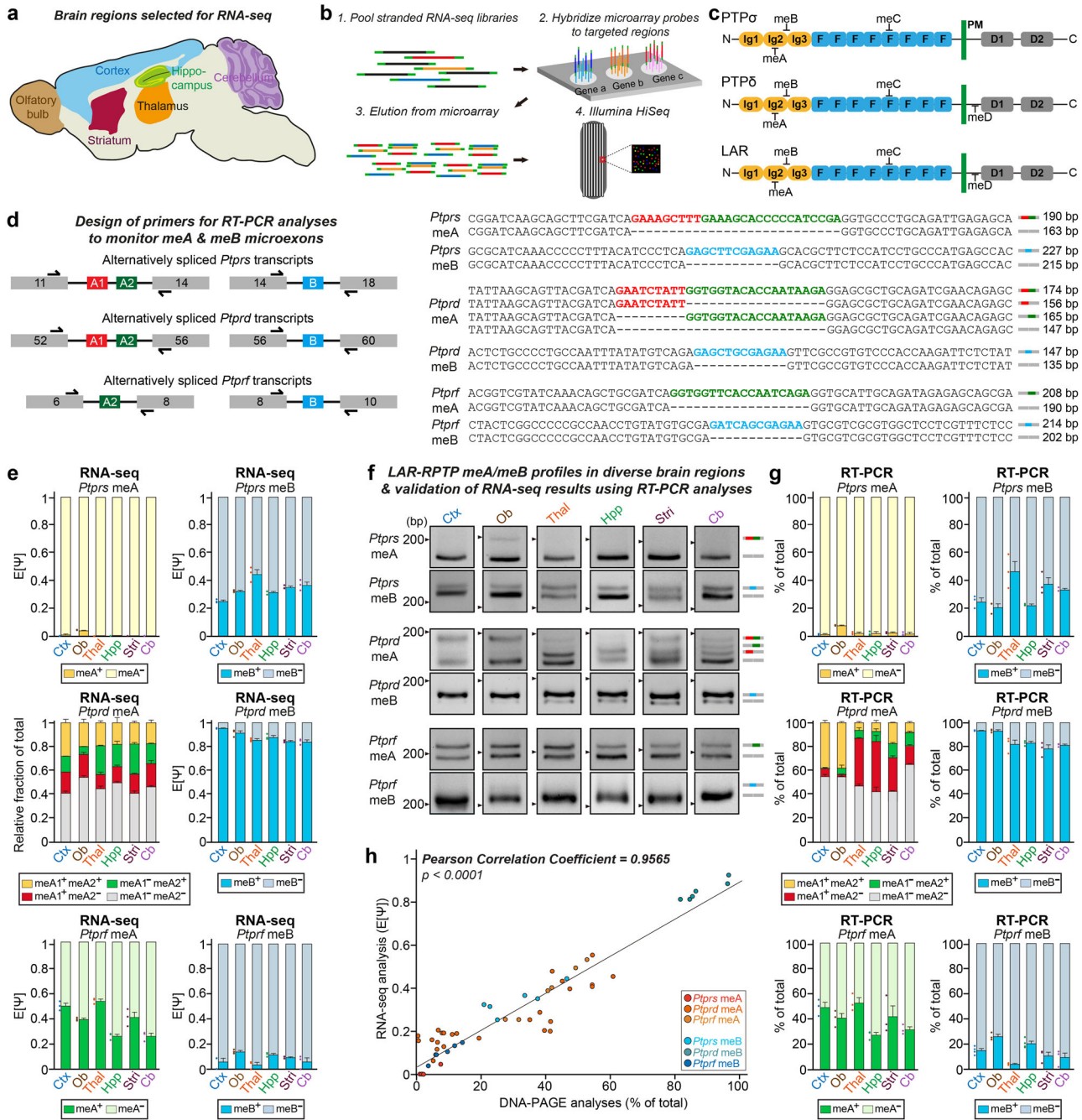

**Fig. 1 | Profiling of LAR-RPTP microexon expression repertoires by targeted deep RNA sequencing. a** Six brain regions chosen for targeted deep RNA-seq analyses. **b** Workflow for targeted deep RNA-seq analyses. The microarray was printed with custom-designed mRNA capture probes for the target genes, and enriched targets were purified for RNA sequencing. **c** Schematic domain structure of three LAR-RPTP family members: PTPσ (encoded by protein tyrosine phosphatase receptor type S polypeptide [PTPRS]); PTPδ (encoded by protein tyrosine phosphatase receptor type D polypeptide [PTPRD]); and LAR (encoded by protein tyrosine phosphatase receptor type F polypeptide [PTPRF]). **d** Structure and inclusion locations of LAR-RPTP microexons. Microexon-flanking PCR probes were designed based on targeted RNA-seq data. PCR products that differed in size by 9 bp were separated using DNA-PAGE. Microexons were identified by sequencing PCR amplicons with each forward primer. **e** Targeted RNA-seq analysis and quantification of expected PSI values (denoted by E[Ψ]) estimated by MAJIQ and relative

fraction of total calculated by Kallisto. Values are expressed as means ± SEMs (*n* = 3 mice for all experimental groups). **f, g** RNA-seq data were confirmed by performing PCR (**f**) on identical RNA samples from six indicated brain regions using microexon-flanking oligonucleotides (see **d**). Quantification (**g**) of percent of total (%) from RT-PCR ('*n*' denotes the number of biological replicates). Values are expressed as means ± SEMs (*n* = 3 mice for all experimental groups; except for *Ptprd* meA [Ctx, Hpp and Stri] and *Ptprf* meB [Ctx and Hpp], *n* = 4 mice). Cb cerebellum, Ctx cortex, Hpp hippocampus, Stri striatum, Ob olfactory bulb, Thal thalamus. **h** Calculation of the Pearson correlation coefficient between the RNA-seq data and the corresponding RT-PCR datasets using the percent of total values. Scatter plot comparison of RNA-seq and qRT-PCR analysis for microexon alternative splicing of LAR-RPTP mRNAs (*n* = 12 for *Ptprs* meA, *Ptprs* meB, *Ptprd* meB, *Ptprf* meA, and *Ptprf* meB; *n* = 48 for *Ptprd* meA; ****p < 0.0001, two-tailed correlation tubular results). Source data are provided as a Source Data File.

abundances in the adult male and female mice (Supplementary Fig. 3). Taken together, these observations determined the expression profiles of two microexons in mRNAs for all three LAR-RPTP, providing a quantitative and reliable method for monitoring specific LAR-RPTP splice variants in vivo.

## Analyses of PTPδ MeA⁺ proteoform expression using targeted proteomics approaches

There is a poor correlation between the transcriptome and proteome (Pearson correlation coefficient≈0.4)[33–35], a noteworthy discordance that is partly explained by post-transcriptional regulation and measurement noise[36,37]. Thus, we sought to determine whether LAR-RPTP proteoforms with or without microexon peptides are expressed and detectable using proteomics approaches. To this end, we performed immunoprecipitation with anti-PTPσ or anti-PTPδ antibodies (see Supplementary Fig. 4a–c for validation of antibodies) using homogenates from the cortex, hippocampus, or striatum of adult mice; homogenates were further digested with trypsin and then subjected to shotgun mass spectrometry analyses (Supplementary Fig. 4d). We found that ionization efficiency was low using this approach owing to the short length of peptides (i.e., 4 amino acids), and the identification of meB peptides for both PTPσ (VAQLR or EAR) and PTPδ (ELR) was ambiguous[38] (Supplementary Fig. 4e, f). Using a bottom-up proteomics approach, we identified PTPδ peptides encoding meA1⁺A2⁺ (SESIGGTPIR), meA1⁻A2⁺ (SGGTPIR), and meA1⁻A2⁻ (SGALQIEQSEESDQGK), but not meA1⁺A2⁻ (SESIGALQIEQSEESDQGK) (Supplementary Fig. 4e); the two reference peptides, NVLELNDVR and VVAVNNIGR, were identified in all PTPδ variants (see Supplementary Fig. 1g). The quality of the identified peptides was validated by calculating cross-correlation score values ($X_{corr}$), with peptides showing $X_{corr}$ values > 1.78 and sequence coverage > 85% being judged as reliable (Supplementary Table 2). However, the PTPσ meA peptide was not detected, consistent with our RT-PCR results (Fig. 1f).

Next, to quantitatively validate our RNA-seq results, we employed parallel reaction monitoring (PRM), an LC−MS-based targeted peptide/protein quantification method that has advantages over traditional proteomic techniques[39–41]. We performed immunoprecipitation on homogenates from the cortex, hippocampus, and striatum of adult mice using anti-PTPδ antibodies, and further subjected homogenates to digestion with trypsin, followed by LC-PRM mass spectrometry analyses after equally spiking stable isotope-labeled peptides of tryptic peptides containing residues encoded by PTPδ meA (Fig. 2a, b). Strikingly, quantitative LC-PRM analyses revealed that the meA1⁺A2⁺ (SESIGGTPIR) peptide, but not other meA peptides, was reproducibly quantified from three biological replicates with a coefficient of variation of 10% or less. One possible explanation for these results is that ionization efficiencies might be lower for peptides other than the meA1⁺A2⁺ (SESIGGTPIR) peptide. For LC-PRM analyses, ratios were calculated by dividing the intensity of the light peptide—an endogenous PTPδ meA1⁺A2⁺ (SESIGGTPIR) peptide—by the intensity of the corresponding heavy peptide, equally spiked for all targeted PTPδ peptides (Fig. 2b). To correct for possible variation in sample enrichment, we normalized these values to the ratios of two reference peptides, calculated in different brain regions (Fig. 2c, d). Expression of the PTPδ meA1⁺A2⁺ variant in the cortex and striatum relatively was greater than that in the hippocampus (Fig. 2d), consistent with the results from RT-PCR analyses (Fig. 2e, f). Despite technical challenges in reliably detecting microexon-encoding peptides in vivo, at minimum, we confirmed that the PTPδ meA1⁺A2⁺ protein variant is differentially expressed across three different brain areas and that DNA-PAGE results, as established in the current study, are invaluable for probing the expression profile of LAR-RPTP microexons.

## Differential usage of LAR-RPTP microexons in distinct cell types across diverse brain areas

Previous RNAscope-based in situ hybridization analyses revealed that LAR-RPTP transcripts are widely expressed in both excitatory and inhibitory neurons in the hippocampus and medial prefrontal cortex (mPFC) of the adult mouse[42]. We first examined whether LAR-RPTP meA/meB microexon expression patterns across diverse brain regions are distinct in different cell types. To profile LAR-RPTP mRNAs in different cell types in five forebrain regions, we crossed the Ai9 reporter mouse line (used to label tdTomato-expressing cells in a Cre activity-dependent manner) with specific Cre lines that drive expression in glutamatergic (Emx1-Cre), GABAergic (Pvalb-Cre and Sst-Cre), and dopaminergic (Drd1-Cre and Drd2-Cre)[43] neuronal populations (Fig. 3a, b).

We discovered that Ptprs meA⁻ variants are prominently expressed in different brain areas regardless of the examined cell type (Fig. 3c–h), in line with previous results (Fig. 1). In contrast, meA inclusions in Ptprd and Ptprf mRNAs were differentially observed in distinct cell types across different brain regions (Fig. 3c–h). Similarly, Ptprs meB⁺ and Ptprs meB⁻ variants were expressed at different ratios in a cell type- and/or brain area-dependent manner. Notably, meB was included at high levels in Ptprd in the examined cell types. Moreover, Ptprf meB⁺ was the principle Ptprf variant expressed, except in cortical pyramidal (Emx1) and interneurons [somatostatin (SST)⁺ and parvalbumin (PV)⁺], olfactory bulb excitatory neurons, thalamic excitatory neurons and SST⁺ interneurons, hippocampal interneurons, and striatal Drd1⁺ dopaminergic neurons. These results suggest that cell-type–specific alternative splicing programs alone do not dictate the distinct regulation of LAR-RPTP microexon usage in mouse neurons.

## Distinct LAR-RPTP microexon splicing in neurons projecting to the same target

A previous study showed that Nrxn transcription signatures are distinct across neural circuits involving the same postsynaptic neuronal populations[44]. Thus, we next asked whether specific patterns of LAR-RPTP microexon expression could be supporting postsynaptic target specificity. To this end, we targeted the convergent projections from two brain regions—the mPFC and hippocampal CA1—using rAAV₂-retro−mediated retrograde tracing[45]. Consistent with results from prior tracing studies[46–48], injection of rAAV₂-retro into the mPFC or dorsal CA1 (dCA1) of adult mice resulted in dense labeling of the respective input neurons at 3 weeks post-injection (Fig. 4a). We then selected a subset of projection inputs (mPFC input regions: cortex, hippocampus, striatum, and thalamus; dCA1 input regions: entorhinal cortex [EC], CA3 and subiculum [SuB]) and performed FACS to isolate GFP-positive neurons (Fig. 4b). RT-PCR analyses revealed that the Ptprs meA splicing signature was roughly similar in the examined input neurons that projected to the mPFC, except for thalamic neurons, which primarily expressed Ptprs meA⁻ variants regardless of their projection identity (Fig. 4c, e). Intriguingly, input neurons projecting to the mPFC from the cortex, hippocampus, or striatum predominantly expressed Ptprs meA⁺ variants (specifically, Ptprs meA1⁺meA2⁻), whereas neurons in the cortex, hippocampus, and striatum that did not project to the mPFC mainly expressed Ptprs meA⁻ variants (Fig. 4c, e; see Fig. 1 for comparison). Ptprd meA variants were expressed at different ratios in distinct mPFC-projecting input neurons that expressed primarily Ptprf meA⁻ variants (Fig. 4c, e). Notably, neurons in the hippocampus and striatum that did not project to mPFC expressed primarily Ptprf meA⁺ variants, but this was not the case for mPFC−non-projecting neurons from the cortex and thalamus. In contrast to the distinct patterns of LAR-RPTP meA splicing signatures, meB inclusion in LAR-RPTP mRNAs appeared to be regulated in a projection-independent manner in mPFC-projecting neuronal populations (Fig. 4c, e). Profiling of

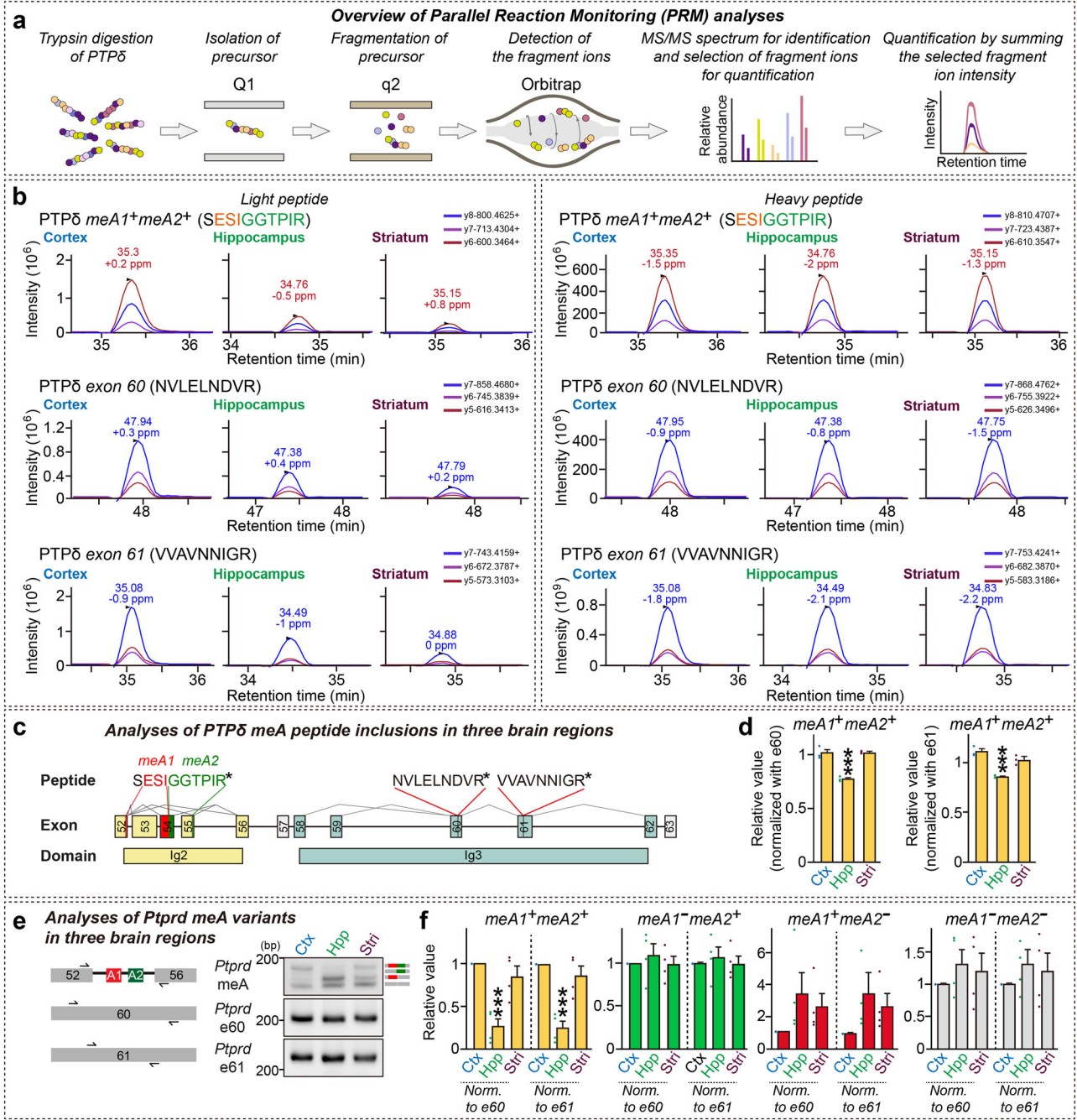

**Fig. 2 | Profiling the PTPδ meA+ proteoform by targeted proteomics.**
**a** Workflow for quantitative Parallel Reaction Monitoring (PRM) mass spectrometry analyses. **b** Chromatograms of three quantitative fragment ions of the endogenous light peptides (left) and their corresponding heavy peptides (right) from PTPδ splice variants containing insertions at both the meA1+ and meA2+ segments. **c** Information on chemically synthesized peptides used for relative quantification of PTPδ meA1+A2+ variants. The position and sequence of peptides within the overall domain structure of PTPδ is indicated. **d** Quantification of PTPδ meA1+A2+ variants in the cortex, hippocampus, and striatum of adult mice using LC-PRM analysis. Peptides encoding *Ptprd* exon 60 (left) or exon 61 (right) were used as controls for normalization. Values are expressed as means ± SEMs ($n = 3$ mice for all experimental groups; ***$p < 0.001$ vs. Ctx; parametric ordinary one-way ANOVA Tukey's multiple comparisons test). Ctx cortex, Hpp hippocampus, Stri striatum, Ig immunoglobulin domain. **e, f** Representative DNA-PAGE gel images (**e**) and quantification (**f**) of the levels of the indicated *Ptprd* meA variant in the cortex, hippocampus, and striatum using RT-PCR. *Ptprd* exon 60 (top) or exon 61 (bottom) was used as a control for normalization. Values are expressed as means ± SEMs ($n = 4$ mice for all experimental groups; ***$p < 0.001$ vs. Ctx; parametric ordinary one-way ANOVA Tukey's multiple comparisons test). Ctx cortex, Hpp hippocampus, Stri striatum. Source data are provided as a Source Data File.

hippocampal neural circuits yielded completely different landscapes of LAR-RPTP microexon expression (Fig. 4d, f). Regardless of projection patterns, *Ptprs* meA− and *Ptprf* meA− variants were primarily detected in the examined input neurons that projected to the dCA1 (Fig. 4d, f). As was the case for mPFC input neurons, *Ptprd* meA variants were expressed at different ratios in distinct dCA1-projecting input neurons

(Fig. 4d, f). Intriguingly, individual *Ptprd* meA+ variants (meA1+meA2+, meA1+meA2− and meA1−meA2−) were differentially expressed in each input neuron in a projection-dependent manner. Unlike the case for mPFC neural circuits, meB inclusions in all three LAR-RPTP mRNAs were regulated in a projection-dependent manner in hippocampal neural circuits (Fig. 4d, f).

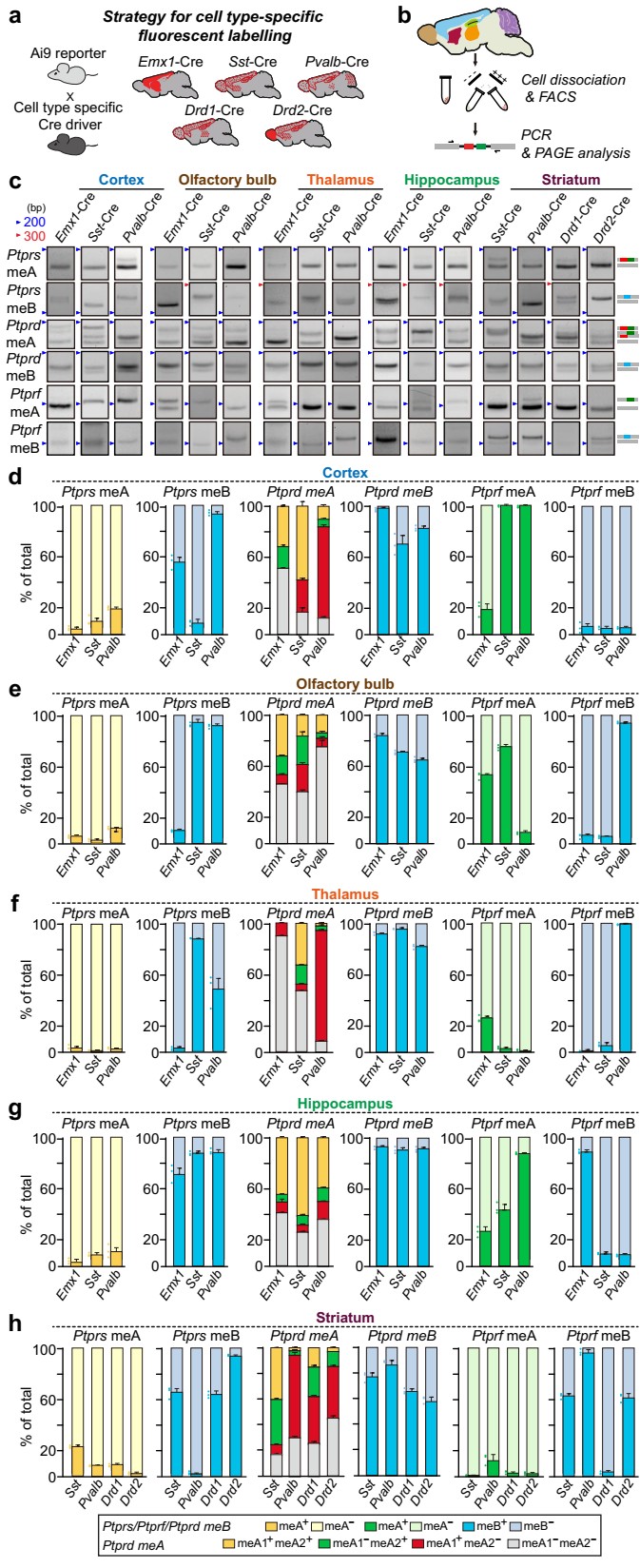

**Fig. 3 | Profiling of cell-type–specific LAR-RPTP microexon expression repertoires. a** Experimental scheme for cell-type–specific fluorescent labeling. Breeding strategy for cell-type labeling using Ai9 reporter mice and the indicated Cre-driver lines, which enable Cre-dependent tdTomato-expression in specified neuronal populations as follows: *Emx1*-Cre for forebrain excitatory neurons; *Sst*-Cre and *Pvalb*-Cre for GABAergic inhibitory neurons; and *Drd1*-Cre and *Drd2*-Cre for dopaminergic neurons. **b** Schematic workflow for FACS and DNA-PAGE analyses of cell-type–specific LAR-RPTP microexon profiling. **c–h** Representative DNA-PAGE (**c**) and quantitative analyses (**d–h**) of LAR-RPTP meA and meB microexon expression repertoires in the indicated cell types from the five targeted brain areas (cortex, olfactory bulb, thalamus, hippocampus, and striatum) of adult male mice. Values are expressed as means ± SEMs (*n* = 3 mice for all experimental groups). Source data are provided as a Source Data File.

receptor for EnvA) was stereotaxically injected into mice of each Cre line, allowing restricted absorption of EnvA-pseudotyped RVs in specific cell types and subsequent retrograde synaptic transport[49]. After validating the specific action of the modified RV system, we separated fluorescently labeled neuronal populations in the main projection inputs of the hippocampal dCA1 by FACS to determine the microexon profiles in each cell type. Both GABAergic interneurons in the dCA1 were synaptically connected to CA3, SuB, and EC regions and exhibited microexon profiles that were similar overall to their region-specific counterparts obtained from whole neuronal populations (Supplementary Fig. 5c, d). These results imply that LAR-RPTP microexon profiles in dCA1 input neurons are indistinguishable, irrespective of the type of efferent target neurons.

## Activity-dependent upregulation of *Ptprd* meA⁺ splice variants in hippocampal dentate gyrus engram cells

Several studies have shown that alternative splicing of Nrxns occurs in an activity-dependent manner[50–54]. Importantly, robust changes in Nrxn1 SS#4 alternative splicing in mice have been reported in response to various forms of neuronal activity, including fear conditioning[50]. We thus asked whether the inclusion of meA or meB in LAR-RPTP mRNAs is also activity-dependent. To this end, we employed the *Fos*-dependent Robust Activity Marking (*F*-RAM) reporter system in conjunction with a fear-conditioning behavioral paradigm[55] to examine whether memory encoding induced alterations in LAR-RPTP meA and/or meB splicing (Fig. 5a). Accordingly, we injected adeno-associated viruses (AAVs) expressing *F*-RAM-mKate2 to selectively label active neuronal ensembles in the adult mouse dentate gyrus (DG), and then provided doxycycline (Dox) in the drinking water for 7 d; Dox was withdrawn 24 h before subjecting mice to fear conditioning (Fig. 5a). Fear memory-activated DG neurons were isolated based on their expression of mKate2 after fear retrieval by fluorescent activated cell sorting (FACS) (Fig. 5a, b). Fear conditioning triggered a significant increase in *Nrxn1* SS#4 inclusion in retrieval-activated neuronal ensembles 24 h after learning compared with that in naïve mice (Fig. 5c, d), as previously reported[50]. Strikingly, there were marked increases in *Ptprd* meA1⁺meA2⁺ and *Ptprf* meA⁺ variants, but not *Ptprs* meA⁺ or any other LAR-RPTP meB⁺ variants (Fig. 5e, f). Note that hippocampal subfields (e.g., CA1 vs. DG) expressed distinct *Ptprd* meA variants, accounting for discrepancies in meA inclusion patterns between crude hippocampal neurons and fear memory engram neurons (see Fig. 1f). Collectively, these results provide evidence that meA inclusion in a subset of LAR-RPTP mRNAs is positively regulated in hippocampal fear memory-activated neuronal populations.

## Differential roles of *Ptprd* meA splicing in regulating distinct modes of excitatory synaptic transmission

We next investigated the physiological significance of LAR-RPTP microexon splicing, particularly in the context of specific neural circuits in vivo. We decided to focus on *Ptprd* meA splicing, based

To test whether differences in LAR-RPTP microexon profiles simply reflect cell-type–specific projection bias based on connectivity to the same target area, we employed a genetically modified rabies virus (RV)-mediated tracing system, in conjunction with specific Cre-driver lines (*Sst*-Cre and *Pvalb*-Cre) (Supplementary Fig. 5a, b). An AAV encoding a Cre recombinase-sensitive TVA receptor (the cognate

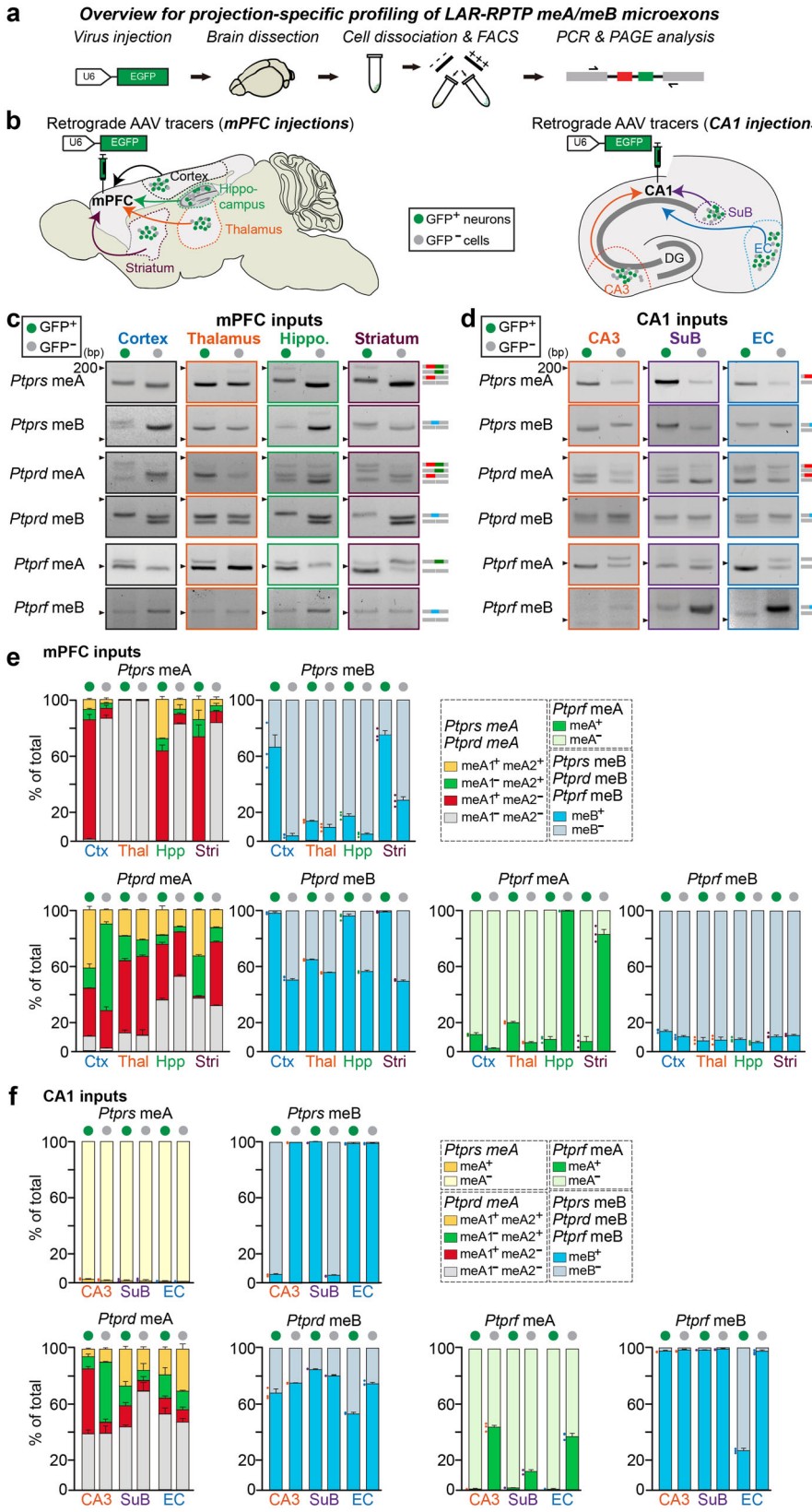

**Fig. 4 | Profiling of circuit-type–specific LAR-RPTP microexon expression repertoires. a** Experimental scheme for projection-specific profiling of LAR-RPTP microexon expression repertoires. **b** Example retrograde labeling of projection neurons with rAAV₂-retro[45]. Retrograde access to the indicated input brain areas projecting to either the mPFC (left) or hippocampal CA1 (right) was assessed 3 wk after delivery of rAAV₂-retro carrying EGFP fluorescence; each input region was

further processed for FACS and DNA-PAGE analyses. DG dentate gyrus, EC entorhinal cortex, SuB subiculum. **c–f** Representative DNA-PAGE (**c, d**) and quantitative analyses (**e, f**) of LAR-RPTP meA and meB profiles in either mPFC-projecting or CA1-projecting neuronal populations. Values are expressed as means ± SEMs ($n = 3$ mice for all experimental groups). Ctx cortex, Thal thalamus, Hpp hippocampus, Stri striatum. Source data are provided as a Source Data File.

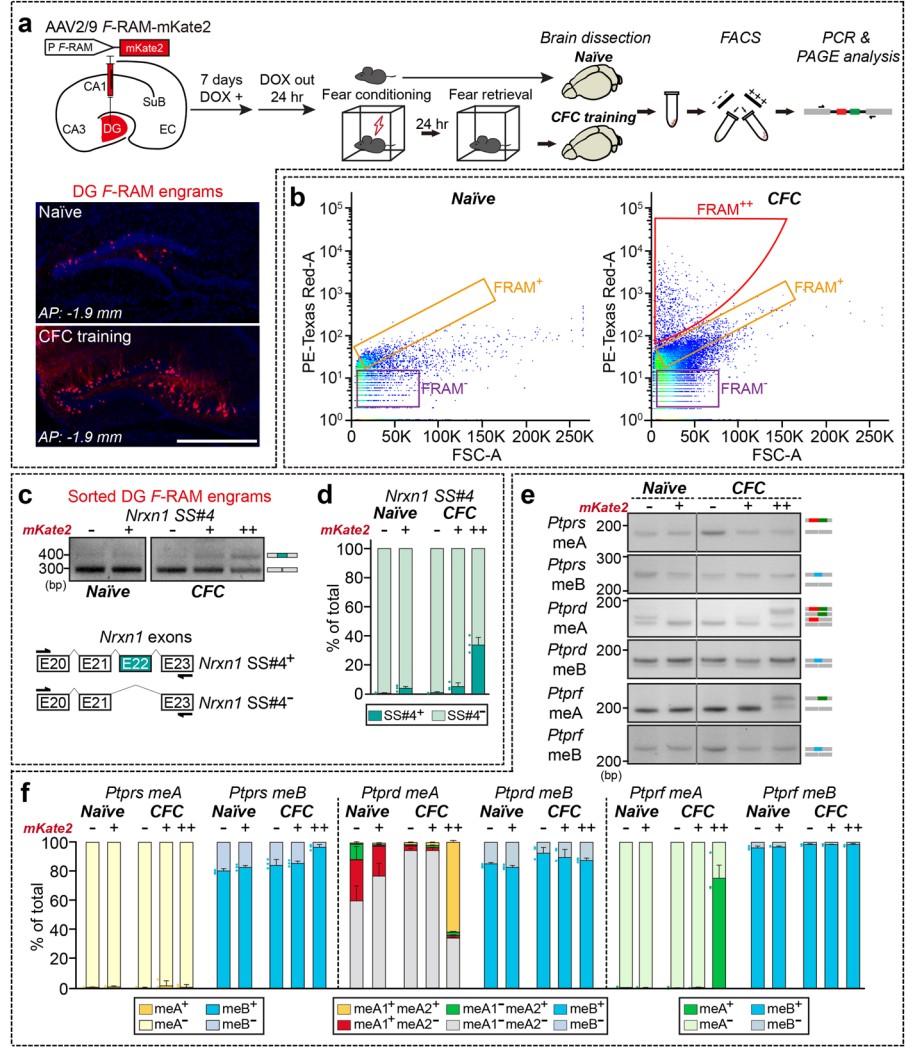

**Fig. 5 | Increased *Ptprd* meA microexon inclusion in fear memory engrams of the adult mouse DG. a** Experimental scheme for labeling and sorting the hippocampal DG engram. Active neuronal ensembles in the mouse dorsal DG were labeled in vivo following contextual fear conditioning (CFC) by stereotaxically injecting adult (~5-week-old) male mice with AAV-*F*-RAM-mKate2. Injected mice were kept in their home cage and administered doxycycline (DOX) for 7 days. Twenty-four hours after DOX withdrawal, mice were subjected to CFC and fear retrieval, after which *F*-RAM⁺ engram populations were sorted by FACS, and LAR-RPTP microexon expression in the DG fear memory engram was profiled by DNA-PAGE analysis. Experimental mice were sacrificed after fear retrieval, whereas naïve mice were kept in their home cages. Representative images of *F*-RAM-mKate2 in the DG are shown. Scale bar = 500 μm. DG dentate gyrus, EC entorhinal cortex, SuB subiculum *F*-RAM Fos-dependent robust activity marking. **b** Representative FACS plots showing the gating strategy for the purification of adult mouse hippocampal DG neurons expressing mKate2 fluorescence induced by the Fos-dependent robust

activity marking (*F*-RAM) system. Neuronal populations expressing mKate2 were low (predominantly *F*-RAM⁻ population) in DG engram neurons from mice reared under home cage conditions (Naïve), but were markedly increased (i.e., *F*-RAM⁺ and *F*-RAM⁺⁺ populations) in DG engram neurons from mice subjected to contextual fear conditioning (CFC). **c, d** Representative DNA-PAGE (**c**) and quantitative analyses (**d**) of *Nrxn1* SS#4 inclusion profiles in the DG fear memory engram. FACS-sorted engrams were further distinguished by mKate2 fluorescence ("+" and "++" depending on intensity); non-engram populations (indicated as "−") were analyzed in parallel. Values are expressed as means ± SEMs (*n* = 3 mice for all experimental groups). **e, f** Representative DNA-PAGE (**e**) and quantitative analyses (**f**) of LAR-RPTP meA and meB profiles in the DG fear memory engram. Note that *Ptprd meA* and *Ptprf meA* levels are significantly greater in F-RAM⁺⁺ engram cells than in non-engram cells. Values are expressed as means ± SEMs (*n* = 3 mice for all experimental groups). Source data are provided as a Source Data File.

on the availability of transgenic mice in which *Ptprd* meA alternative splicing (herein termed *Ptprd* meA-cKO) could be conditionally modulated[24] and the observation that *Ptprd* meA alternative splicing events reflect neuronal activity levels (Fig. 5). We stereotactically injected AAVs expressing active Cre recombinase (AAV-Cre) or inactive Cre recombinase (AAV-ΔCre; control) into the cortex of *Ptprd* meA-cKO mice and analyzed the profiles of *Ptprd* meA microexons (Supplementary Fig. 6a). In keeping with our previous results (Fig. 1), cortical cells infected with AAV-ΔCre predominantly expressed *Ptprd* meA1⁺meA2⁺, meA1⁺meA2⁻ and meA1⁻meA2⁻ variants (Supplementary Fig. 6a). Injection of AAV-Cre successfully eliminated *Ptprd* meA1⁺meA2⁺,

meA1⁻meA2⁺ variants, but not meA1⁺meA2⁻ and meA1⁻meA2⁻ variants, validating the *Ptprd* meA-cKO mouse line.

To first elucidate the effect of circuit-specific deletion of PTPδ on synaptic function, we injected *Ptprd* floxed mice into the dCA1 with a *trans*-neuronally transported version of Flpo fused to wheat-germ agglutinin (WGA-Flpo), and into the CA3, SuB, and entorhinal cortex (EC) with an Flp-dependent AAVs expressing Cre recombinase (AAV-fDIO-Cre) (Fig. 6a). We confirmed that PTPδ protein was expressed in the CA3, SuB and EC using a PTPδ-tdTomato reporter mouse line[24] (Supplementary Fig. 6b, c). We then performed electrophysiological recordings to measure synchronous evoked excitatory postsynaptic currents (eEPSCs) of hippocampal dCA1 neurons by

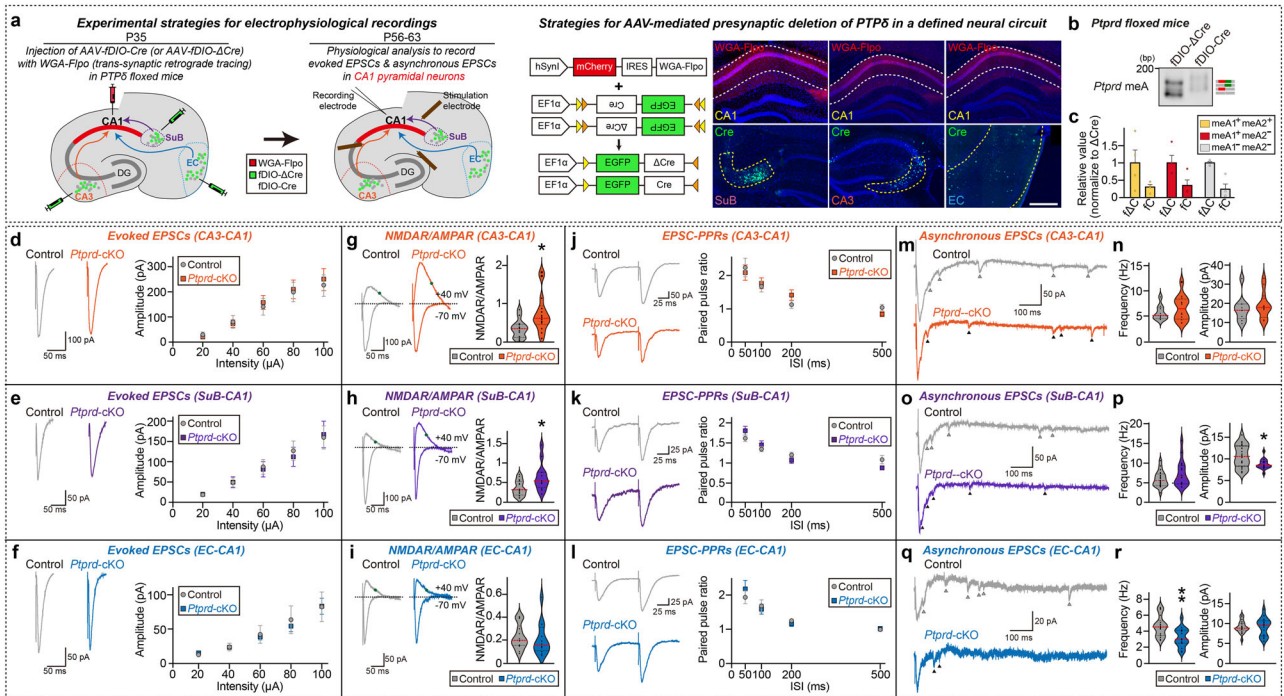

**Fig. 6 | Effects of neural circuit-specific deletion of PTPδ on the specific glutamatergic synaptic properties of hippocampal CA1 pyramidal neurons.**
**a** Schematic depiction of the experimental design for in vivo manipulations of presynaptic PTPδ in three different neural circuits encompassing hippocampal CA1 neurons. Representative immunofluorescence images showing fDIO-ΔCre or fDIO-Cre expression in SuB, CA3 or EC neurons. Scale bar, 1 mm. **b, c** Representative DNA-PAGE (**b**) and quantitative analyses (**c**) of *Ptprd* meA profiles in CA1-projecting subicular neurons expressing fDIO-ΔCre or fDIO-Cre of PTPδ[f/f] mice. Values are expressed as means ± SEMs (*n* = 4 mice for all experimental groups). fΔC fDIO-ΔCre, fC fDIO-Cre. **d–f** Whole-cell recordings of eEPSCs from CA3–dCA1, SuB–dCA1, and EC–dCA1 synapses. Representative traces and average eEPSC I-O curve for CA3– dCA1 ('*n*' denotes number of cells/mice; **d** Control, *n* = 13/5; *Ptprd*-cKO, *n* = 14/5), SuB–dCA1 (**e** Control, *n* = 16/5; *Ptprd*-cKO, *n* = 12/5), and EC–dCA1 synapses (**f** Control, *n* = 14/5; *Ptprd*-cKO, *n* = 12/5). Data are presented as means ± SEMs. **g–i** Measurement of NMDAR/AMPAR eEPSCs from CA3–dCA1, SuB–dCA1, and EC–dCA1 synapses. Representative traces and average NMDAR/

AMPAR eEPSCs for CA3–dCA1 ('*n*' denotes number of cells/mice; **g** Control, *n* = 13/5; *Ptprd*-cKO, *n* = 14/5), SuB–dCA1 (**h** Control, *n* = 16/5; *Ptprd*-cKO, *n* = 12/5), and EC–dCA1 synapses (**i** Control, *n* = 14/5; *Ptprd*-cKO, *n* = 12/5). Data are presented as means ± SEMs (**p* < 0.05; two-tailed non-parametric Mann−Whitney *U*-test).
**j–l** Measurement of paired-pulse ratio of eEPSCs from CA3–dCA1, SuB–dCA1, and EC–dCA1 synapses. Representative traces and average PPR for CA3–dCA1 ('*n*' denotes number of cells/mice; **j** Control, *n* = 13/5; *Ptprd*-cKO, *n* = 14/5), SuB–dCA1 (**k** Control, *n* = 16/5; *Ptprd*-cKO, *n* = 12/5), and EC–dCA1 synapses (**l** Control, *n* = 14/5; *Ptprd*-cKO, *n* = 12/5). Data are presented as means ± SEMs. **m–r** Whole-cell recordings of aEPSCs from CA3–dCA1, SuB–dCA1, and EC–dCA1 synapses. Representative traces and average aEPSCs for CA3–dCA1 ('*n*' denotes number of cells/ mice; **m, n** Control, *n* = 9/4; *Ptprd*-cKO, *n* = 12/5), SuB–dCA1 (**o, p** Control, *n* = 20/5; *Ptprd*-cKO, *n* = 16/5), and EC–dCA1 synapses (**q, r** Control, *n* = 18/5; *Ptprd*-cKO, *n* = 16/5). Data are presented as means ± SEMs (**p* < 0.05, ***p* < 0.01; two-tailed non-parametric Mann−Whitney *U*-test). Source data are provided as a Source Data File.

stimulating fibers of the Schaffer collateral (SC) pathway (i.e., CA3→dCA1), SuB→dCA1 circuit, and the temporoammonic (TA) pathway (i.e., EC→dCA1). Presynaptic deletion of PTPδ at CA3→dCA1, SuB→dCA1, and EC→dCA1 circuits (verified by concomitant DNA-PAGE analyses; Fig. 6b, c) did not alter the amplitudes of α-amino-3-hydroxy-5-methyl-4-isoxazolepropionic acid receptor (AMPAR)-mediated eEPSCs (Fig. 6d–f). In contrast, *N*-methyl-D-aspartate receptor (NMDAR)-mediated postsynaptic responses were increased in dCA1 pyramidal neurons innervated by PTPδ-deleted CA3 or SuB neurons, but not those innervated by PTPδ-deleted EC neurons (Fig. 6g–i). There were no changes in release probability in any dCA1 pyramidal neurons innervated by PTPδ-deficient input neurons, as assessed by measuring paired-pulse ratio (PPR) (Fig. 6j–l). Considered in light of a recent report that asynchronous release sites are spatially linked to NMDAR clusters, the specific regulation of NMDAR-mediated postsynaptic responses by PTPδ at certain neural circuits prompted us to examine whether PTPδ is also involved in the regulation of asynchronous release[56]. Strikingly, presynaptic ablation of PTPδ at different hippocampal neural circuits produced distinct forms of alterations in asynchronous EPSCs (aEPSCs): no changes in aEPSC frequency or amplitude (CA3→dCA1), decreased aEPSC amplitude (SuB→dCA1), and decreased aEPSC frequency (EC→dCA1) (Fig. 6m–r). These data suggest that PTPδ is involved in differentially conferring quantal

properties of excitatory synaptic transmission in response to the distinct inputs of hippocampal dCA1 pyramidal neurons. Because PTPδ deletion at the examined circuits did not alter PPRs, changes in functional synapse number or synaptic vesicle organization might underlie the changes in aEPSCs in SuB→dCA1 and EC→dCA1 circuits.

We next asked whether the circuit-selective loss of PTPδ is recapitulated by the deletion of PTPδ meA[+] variants in the respective neural circuit (Fig. 7a). To test this, we employed *Ptprd* meA floxed mice for virus injections of WGA-Flpo into the dCA1 and AAV-fDIO-Cre into the EC, CA3, and SuB, and performed the same electrophysiological recordings. Presynaptic deletion of *Ptprd* meA[+] variants at CA3→dCA1, SuB→dCA1, and EC→dCA1 circuits (verified by concomitant DNA-PAGE analyses; Fig. 7b) did not alter the amplitudes of AMPAR-mediated eEPSCs (Fig. 7c–e), an outcome reminiscent of *Ptprd*-cKO phenotypes (Fig. 6d–f). Remarkably, deletion of PTPδ meA[+] variants in CA3 neurons did not alter NMDAR-mediated postsynaptic responses in innervated dCA1 neurons, whereas deletion of PTPδ meA[+] variants in SuB neurons increased these responses in innervated dCA1 neurons (Fig. 7f–h), suggesting that PTPδ meA[+] variants at the SuB→dCA1 circuit, but not the CA3→dCA1 circuit, are critical for regulating NMDAR-mediated postsynaptic responses. Moreover, regardless of which hippocampal dCA1 circuit was examined, PTPδ meA[+] variants are not required for dictating aEPSCs (Fig. 7i–n). These results suggest that

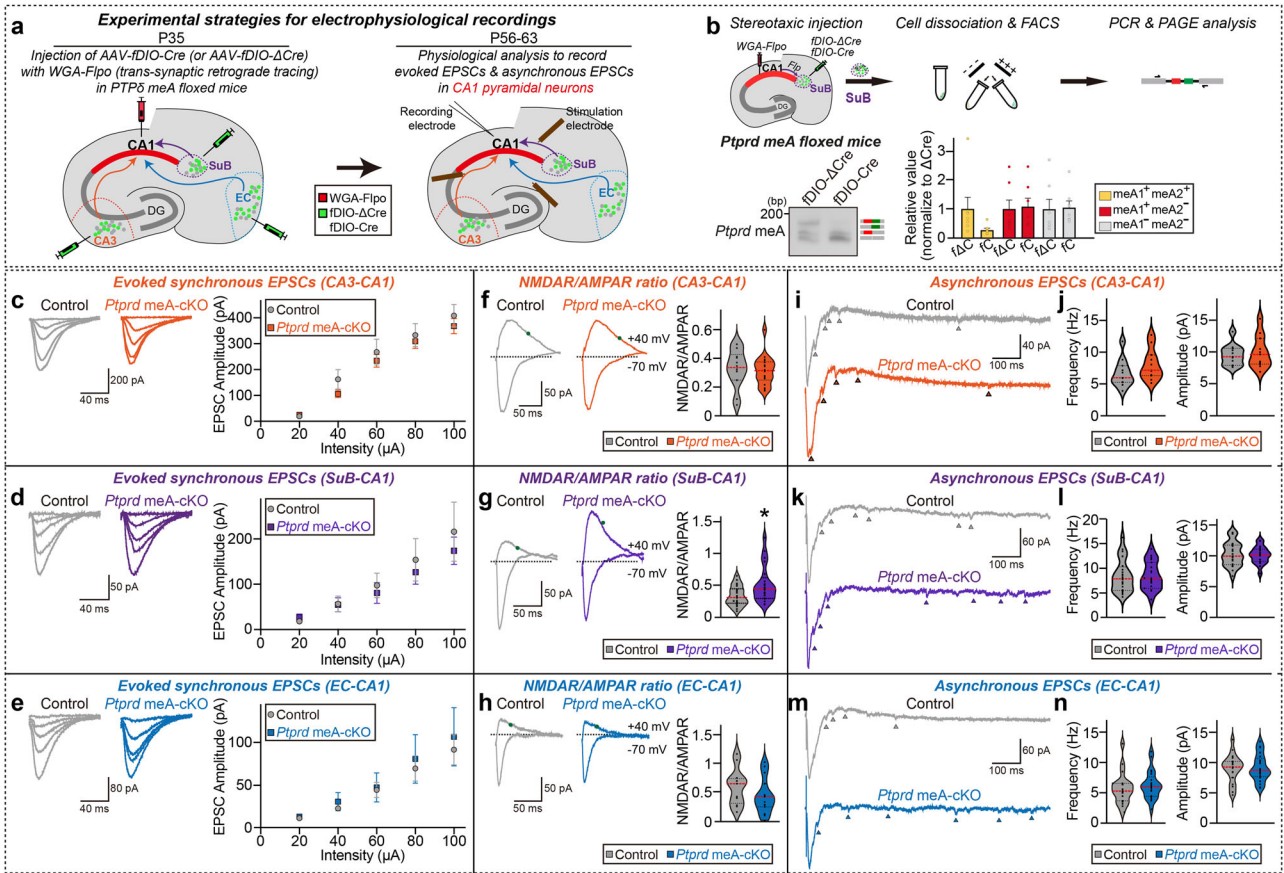

**Fig. 7 | Presynaptic PTPδ employs meA Microexon inclusion for differential regulation of distinct excitatory synaptic transmission in hippocampal CA1 neural circuits. a** Schematic depiction of the design for in vivo manipulations of presynaptic PTPδ meA variants in three different neural circuits encompassing hippocampal CA1 neurons. The hippocampal CA1 of PTPδ meA floxed mice was injected with AAV WGA-Flpo, after which the three indicated CA1 input regions were infected with fDIO-ΔCre or fDIO-Cre at P35. Acute slices were analyzed at P56–P63 by stimulating presynaptic fibers from the CA3 (orange), entorhinal cortex (EC; blue), or subiculum (SuB; violet), and monitoring postsynaptic responses in CA1 pyramidal neurons. **b** Workflow of stereotaxic injections, FACS, and DNA-PAGE analyses (top). Representative DNA-PAGE and quantitative analyses (bottom) of *Ptprd* meA profiles in CA1-projecting subicular neurons expressing fDIO-ΔCre or fDIO-Cre of PTPδ meA floxed mice. Values are expressed as means ± SEMs (*n* = 7 mice for all experimental groups). fΔC fDIO-ΔCre, fC fDIO-Cre. **c–e** Whole-cell recordings of eEPSCs from CA3–dCA1, SuB–dCA1, and EC–dCA1 synapses. Representative traces and average eEPSC I-O curve for CA3–dCA1 ('*n*' denotes number of cells/mice; **c** Control, *n* = 14/5; *Ptprd* meA-cKO, *n* = 18/5), SuB–dCA1 (**d** Control, *n* = 22/6; *Ptprd* meA-cKO, *n* = 21/6), and EC–dCA1 synapses (**e** Control, *n* = 12/5; *Ptprd* meA-cKO, *n* = 15/5). Data are presented as means ± SEMs. **f–h** Measurement of NMDAR/AMPAR eEPSCs from CA3–dCA1, SuB–dCA1, and EC–dCA1 synapses. Representative traces and average NMDAR/AMPAR eEPSCs for CA3–dCA1 ('*n*' denotes number of cells/mice; **f,** Control, *n* = 14/5; *Ptprd* meA-cKO, *n* = 18/5), SuB–dCA1 (**g** Control, *n* = 22/6; *Ptprd* meA-cKO, *n* = 21/6), and EC–dCA1 synapses (**h** Control, *n* = 12/5; *Ptprd* meA-cKO, *n* = 15/5). Data are presented as means ± SEMs (*p < 0.05; two-tailed non-parametric Mann–Whitney *U*-test). **i–n** Whole-cell recordings of aEPSCs from CA3–dCA1, SuB–dCA1, and EC–dCA1 synapses. Representative traces and average aEPSCs for CA3–dCA1 ('*n*' denotes number of cells/mice; **i, j** Control, *n* = 11/4; *Ptprd* meA-cKO, *n* = 11/4), SuB–dCA1 (**k, l** Control, *n* = 24/6; *Ptprd* meA-cKO, *n* = 19/6), and EC–dCA1 synapses (**m, n** Control, *n* = 19/6; *Ptprd* meA-cKO, *n* = 21/6). Data are presented as means ± SEMs. Source data are provided as a Source Data File.

*Ptprd* meA splicing plays a critical role in regulating specific synapse properties in a circuit context-dependent manner.

Electron microscopic analyses using a PTPδ-tdTomato reporter line showed predominant localization of PTPδ-tdTomato protein at glutamatergic axon terminals[24], whereas stimulated emission depletion microscopy revealed the nanoscale organization of endogenous PTPδ at both glutamatergic and GABAergic axon terminals[57]. Thus, to test whether loss of PTPδ in GABAergic interneurons also impairs synaptic properties, we delivered AAV-fDIO-Cre-GFP or AAV-fDIO-ΔCre-GFP into the hippocampal dCA1 of PV-Flp::*Ptprd* floxed or SST-Flp::*Ptprd* floxed mice (Supplementary Fig. 7a). Three weeks post-injection, we identified PV+ or SST+ neurons expressing both AAVs in the hippocampal dCA1 area and measured evoked inhibitory post-synaptic currents (eIPSCs) in the pyramidal neurons in acute brain slices (Supplementary Fig. 7b–k). We found that the eIPSC amplitudes were not different between AAV-fDIO-Cre–infected SST or PV neurons and AAV-fDIO-ΔCre-infected SST or PV neurons, suggesting that PTPδ

expressed in GABAergic interneurons in the dCA1 area is not functionally required for inhibitory synaptic strength in the hippocampus.

## *Ptprd* meA splicing in subicular neurons dictates object-location memory

Lastly, we tested whether PTPδ deletion in a specific neural circuit impacts mouse behaviors and determined whether these resulting behavioral outcomes are caused by the presence of PTPδ meA+ variants. Accordingly, we injected *Ptprd* floxed mice in the dCA1 with WGA-Flpo and the CA3, SuB, and EC with AAV-fDIO-Cre or AAV-fDIO-ΔCre, and performed a battery of behavioral tests 4 weeks after injections, including open field (OF), novel object recognition, novel object-location, three-chamber, and prepulse inhibition (PPI) tests, to first determine whether hippocampal dCA1 electrophysiological phenotypes in the respective circuit-specific *Ptprd* conditional KO (cKO) mice are linked to behavioral abnormalities (Fig. 8a, upper). We found that mice with CA3→dCA1 circuit-specific *Ptprd*-cKO or SuB→dCA1 circuit-specific

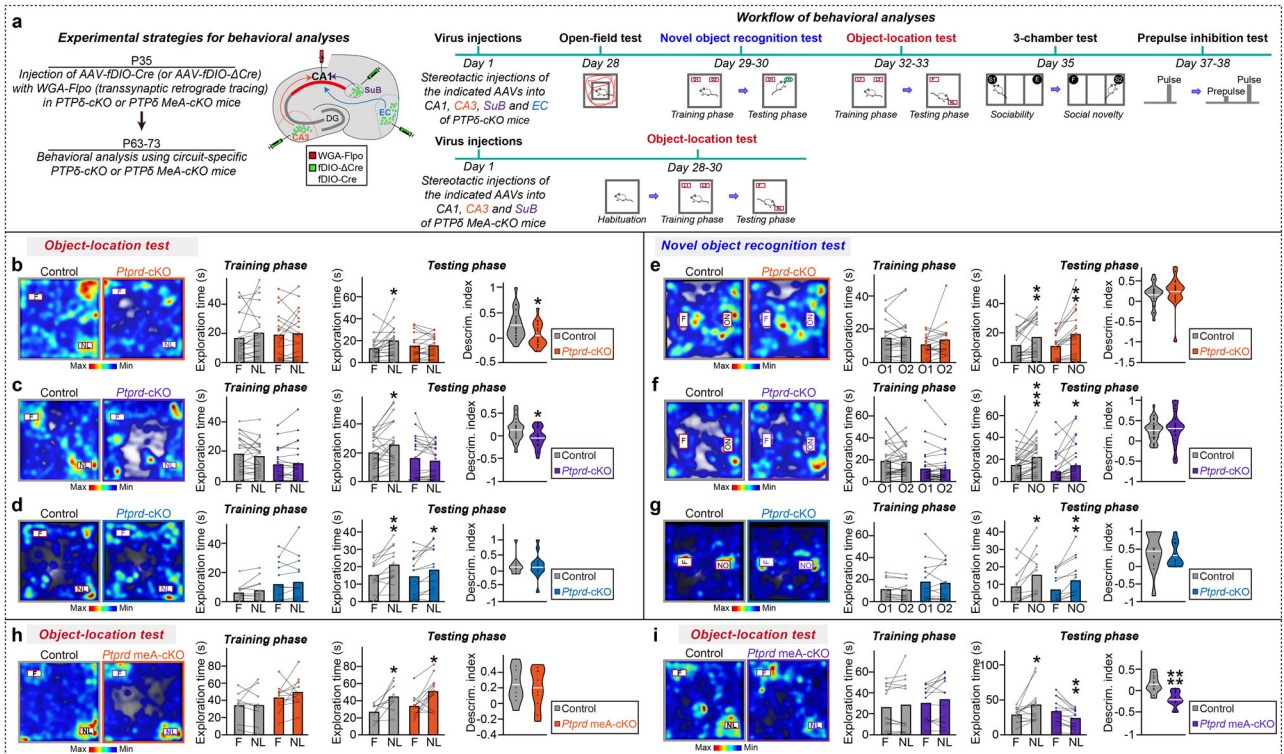

**Fig. 8 | Presynaptic ablation of PTPδ or PTPδ meA⁺ variants in subicular neurons impairs object-location memory in mice. a** Schematic diagram (left) depicting behavioral analyses and test sequence of behavioral analyses using *Ptprd*-cKO or *Ptprd* meA-cKO (right). DG dentate gyrus, EC entorhinal cortex, SuB subiculum. **b–d** Representative heat maps of the object-location test in CA3→ dCA1 circuit-specific *Ptprd*-cKO (**b** left), SuB→CA1 circuit-specific *Ptprd*-cKO (**c** left), EC→ dCA1 circuit-specific *Ptprd*-cKO (**d** left) mice, and summary graphs (right). Data are presented as means ± SEMs ('*n*' denotes number of mice; Control, *n* = 21; *Ptprd*-cKO, *n* = 18 for **b**; Control, *n* = 22; *Ptprd*-cKO, *n* = 21 for **c**; Control, *n* = 12; *Ptprd*-cKO, *n* = 14 for **d**; \**p* < 0.05, \*\**p* < 0.01; two-tailed paired *t*-test [exploration time] or two-tailed non-parametric Mann–Whitney *U*-test [discrimination index]). F familiar location, NL novel location. **e–g** Representative heat maps of the novel object-recognition memory test in CA3→dCA1 circuit-specific *Ptprd*-cKO (**e** left), SuB→ dCA1 circuit-specific *Ptprd*-cKO (**f** left), EC→dCA1 circuit-specific *Ptprd*-cKO (**g** left) mice, and summary graphs (right). Data are presented as means ± SEMs ('*n*'

denotes number of mice; Control, *n* = 21; *Ptprd*-cKO, *n* = 18 for **e**; Control, *n* = 30; *Ptprd*-cKO, *n* = 31 for **f**; Control, *n* = 12; *Ptprd*-cKO, *n* = 13 for **g**; \**p* < 0.05, \*\**p* < 0.01, \*\*\**p* < 0.001; two-tailed paired *t*-test [exploration time] or two-tailed non-parametric Mann–Whitney *U*-test [discrimination index]). O object, F familiar object, NO novel object. **h** Representative heat maps of the object-location test in CA3→ dCA1 circuit-specific *Ptprd* meA-cKO (left) mice, and summary graphs (right). Data are presented as means ± SEMs ('*n*' denotes number of mice; Control, *n* = 9; *Ptprd* meA-cKO, *n* = 11; \**p* < 0.05; two-tailed paired t-test [exploration time] or two-tailed non-parametric Mann–Whitney *U*-test [discrimination index]). **i** Representative heat maps of the object-location test in SuB→ dCA1 circuit-specific *Ptprd* meA-cKO (left). Summary graphs (right). Data are presented as means ± SEMs ('*n*' denotes number of mice; Control, *n* = 13; *Ptprd* meA-cKO, *n* = 13; \**p* < 0.05, \*\**p* < 0.01, \*\*\*\**p* < 0.0001; two-tailed paired *t*-test [exploration time] or two-tailed non-parametric Mann–Whitney *U*-test [discrimination index]). Source data are provided as a Source Data File.

*Ptprd*-cKO, but not EC→dCA1 circuit-specific *Ptprd*-cKO, exhibited impaired object-location memory (Fig. 8b–d), in line with previous reports implicating these circuits in object-place memory[47,58]. In contrast, all three dCA1 circuit-specific *Ptprd*-cKO mice showed normal locomotor activity (as assessed by the OF test; Supplementary Fig. 8a, b, g, h, m, n), normal acoustic startle response to a sudden intense stimulus (as assessed by the PPI test; Supplementary Fig. 8c, i, o), normal sociability and social novelty recognition memory (as assessed by the three-chamber test; Supplementary Fig. 8d–f, 8j–l and 8p–r), and normal novel object-recognition memory (Fig. 8e–g).

To examine whether deletion of PTPδ meA⁺ variants in either the CA3→dCA1 or SuB→dCA1 circuit recapitulates the decreased object-location memory shown in the respective circuit-specific *Ptprd*-cKO mice, we injected *Ptprd* meA floxed mice with WGA-Flpo into the dCA1 and with AAV-fDIO-Cre or AAV-fDIO-ΔCre into the CA3 or SuB and performed object-location memory tests (Fig. 8a, lower). Notably, CA3→dCA1 circuit-specific *Ptprd* meA-cKO mice exhibited object-location memory comparable to that of control mice; in contrast, SuB→dCA1 circuit-specific *Ptprd* meA-cKO mice displayed impaired object-location memory (Fig. 8h, i), suggesting that *Ptprd* meA⁺ variants expressed in SuB neurons, but CA3 or EC neurons, that project to

dCA1 neurons are required for mediating proper object-location memory in mice.

To determine if the activity-triggered PTPδ meA splicing alterations could directly regulate excitatory synaptic properties and behavior, we generated designer receptors that were exclusively activated by designer drug (DREADD)-based chemogenetics (Fig. 9a). We injected wild-type mice with Cre-dependent inverted open reading frame (DIO) AAVs expressing excitatory (hM3Dq) DREADDs (AAV-DIO-hM3Dq) or inhibitory DREADDs (AAV-DIO-hM4Di), and then administered the recently developed designer ligand, deschloroclozapine (DCZ)[59] or saline (SA), via intraperitoneal injection (100 µg/kg) 24 h prior to beginning the electrophysiological recordings or behavioral experiments (Fig. 9a). We validated the DCZ-induced excitation or inhibition of DREADD receptors by measuring firing rates in SuB pyramidal neurons (Fig. 9b, c). SuB neurons expressing hM3Dq and hM4Di exhibited increased and decreased firing rates, respectively, after application of DCZ to brain sections (Fig. 9b, c). We next examined whether altered excitability in SuB neurons projected to dCA1 could alter the *Ptprd* meA profile. Injection of AAV-DIO-hM3Dq or AAV-DIO-hM4Di with AAV-fDIO-Cre into SuB or with WGA-Flpo into the dCA1 of adult mice, followed by the administration of DCZ or SA, was found to

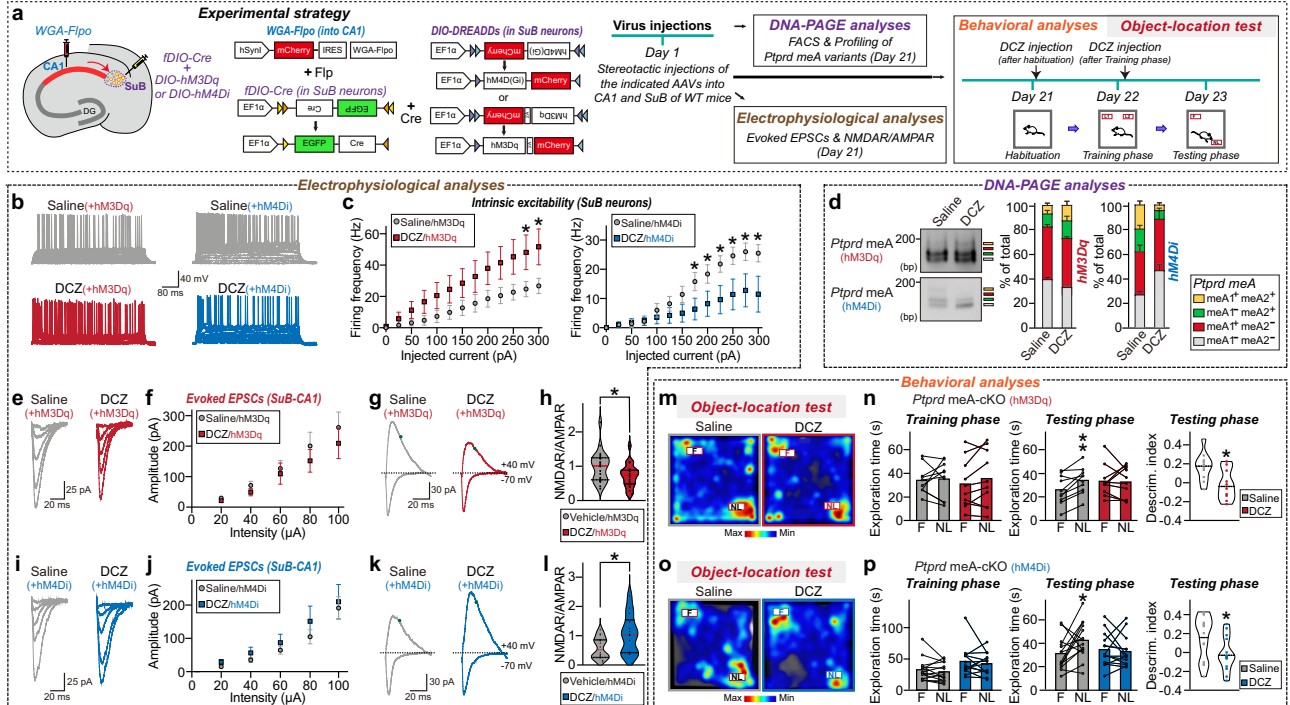

**Fig. 9 | Activity-triggered SuB→dCA1 circuit-specific changes of PTPδ meA insertion modulates NMDA receptor-mediated responses and object-location memory in mice. a** Schematic of strategy used to assess how SuB-dCA1 circuit-specific activation impacted PTPδ meA profiles, NMDA receptor-mediated post-synaptic responses, and object-location memory. Abbreviations: DG, dentate gyrus; DREADD, designer receptors exclusively activated by designer drugs; SuB, sub-iculum. **b, c** Representative traces (**b**) of action potential firing in SuB pyramidal neurons exposed to current injections, and quantification thereof (**c**). Data were acquired from vehicle-treated (gray) or DCZ-treated slices expressing hM3Dq (red) or hM4Di (blue). Data are presented as means ± SEMs ('n' denotes number of cells/mice; vehicle/hM3Dq, $n = 11/3$; DCZ/hM3Dq, $n = 11/3$; vehicle/hM4Di, $n = 10/3$; DCZ/hM4Di, $n = 10/3$; *$p < 0.05$; two-tailed non-parametric Mann−Whitney $U$ test). DCZ deschlorochlozapine. **d** Representative DNA-PAGE (left) and quantitative analyses (right) of *Ptprd* meA microexon expression repertoires in SuB neurons of adult male mice. Values are expressed as means ± SEMs (Saline/hM3Dq, $n = 7$ mice; DCZ/hM3Dq, $n = 7$ mice; Saline/hM4Di, $n = 5$ mice; DCZ/hM4Di, $n = 5$ mice). **e–l** Whole-

cell recordings of eEPSCs from SuB-dCA1 synapses. Representative traces (**e, i**) of AMPAR-EPSC I-O curves at the SuB→dCA1 circuit, acquired from hM3Dq- or hM4Di-expressing slices treated with vehicle alone (saline; gray) or with DCZ (red or blue), and average of AMPAR-EPSC amplitudes plotted as a function of stimulus intensity (**f, j**). Representative traces (**g, k**) of NMDAR-EPSC/AMPAR-EPSC ratio, acquired from hM3Dq- or hM4Di-expressing slices treated with vehicle alone (saline; gray) or DCZ (red or blue), and violin plots (**h, l**). Data are presented as means ± SEMs ('n' denotes number of cells/mice; vehicle/hM3Dq, $n = 21/5$; DCZ/hM3Dq, $n = 23/5$; vehicle/hM4Di, $n = 16/5$; DCZ/hM4Di, $n = 14/5$; *$p < 0.05$; two-tailed non-parametric Mann−Whitney $U$ test). **m–p** Representative heat maps of object-location test results obtained from WT mice with DCZ-mediated hM3Dq excitation or hM4Di inhibition of subicular neurons (**m, o**) and summary graphs (**n, p**). Data are presented as means ± SEMs (hM3Dq: saline, $n = 10$ mice; DCZ, $n = 10$ mice; hM4Di: saline, $n = 14$ mice; DCZ, $n = 14$ mice; *$p < 0.05$, **$p < 0.01$; two-tailed paired $t$-test [exploration time] or two-tailed Mann−Whitney $U$-test [discrimination index]). Source data are provided as a Source Data File.

increase and decrease the amount of *Ptprd* meA+ variants, respectively (Fig. 9a, d). Consistent with the results described earlier (Figs. 5, 7g), NMDAR-mediated postsynaptic responses were decreased in dCA1 connected to chemogenetically activated SuB neurons but increased in dCA1 connected to chemogenetically inhibited SuB neurons (Fig. 9g, h, k, l). Remarkably, both SuB→dCA1 circuit-specific chemogenetic bidirectional manipulations impaired the object-location memory of the mice (Fig. 9m–p). These data support the notion that presynaptic PTPδ meA splicing at the SuB→dCA1 circuit might govern proper gating of postsynaptic NMDAR function in controlling object-location memory.

To exclude the possibility that the absolute number of presynaptic inputs to dCA1 dictates the requirement for a specific circuit in mediating object-location memory, we again performed retrograde tracing experiments by injecting rAAV₂-retro into the dCA1 area. Three weeks after injections, we confirmed that, although dCA1 neurons are connected to the examined input regions, they are innervated primarily from the CA3 rather than the SuB or EC (Supplementary Fig. 6d, e); this is in line with the previous result[60]. We also observed that ablation of presynaptic PTPδ meA+ variants in SuB neurons that project to dCA1 neurons did not significantly alter the connection these neurons (Supplementary Fig. 9). Our results suggest that changes in the NMDAR-mediated postsynaptic currents and object-location memory

of SuB→dCA1 circuit-specific *Ptprd* meA-cKO mice are unlikely to be due to compromised anatomical connectivity between SuB and dCA1. Overall, our results strongly suggest that greater connectivity between different brain regions does not, in and of itself, determine the strength and properties of the connected synapses and dictate the manifestation of a specific behavior.

## Discussion

Alternative mRNA splicing of synaptic cell-adhesion molecules (CAMs) has emerged as a key mechanism for instructing synapse formation and function. A flurry of recent studies has identified numerous microexons among neural-specific alternative exons implicated in nervous system development and disorders[19,21,26,61]. In addition, a subset of microexons mediates animal behaviors[62,63]. However, the precise functions of most microexons in the context of diverse synapse types and neural circuits remain enigmatic[64,65]. In the present study, we employed targeted RNA-seq with sufficient coverage depth to illuminate spatiotemporal expression profiles and differential splicing of two microexons—meA and meB—of LAR-RPTP mRNAs. Previous RNA-seq studies employing whole-genome sequencing approaches did not convincingly determine the expression of these LAR-RPTP microexons because they lacked the higher sequencing read depth (as adopted in the current study) required for microexon splicing analysis.

We then exploited the high resolving power of polyacrylamide gel electrophoresis to validate the targeted RNA deep sequencing results using quantitative PCR, in conjunction with Sanger sequencing. This toolkit allowed us to elucidate physiological significance in the context of specific brain regions, cell types, and neural circuits. Importantly, PRM mass spectrometry quantification of PTPδ splice variants containing or lacking meA revealed that the abundance of each PTPδ mRNA splice isoform with/without meA is compatible with that of the corresponding PTPδ proteoforms in three different brain areas, verifying the reliability of our microexon profiling. In addition, these experiments are the first to address the putative role of LAR-RPTP microexons as 'regulatory adhesion codes' that instruct context-dependent organization of various synaptic pathways, including the connectivity of specific neuronal populations, the specific synaptic strength and molecular composition of these connections, and their propensity to be modified by external cues. We found that meA/meB microexons of LAR-RPTPs exhibit differential expression patterns, even in the same cell type residing in different brain regions (depicted in Supplementary Table 5). In addition, neurons with common long-range projection targets or cell-type–specific connectivity do not necessarily employ similar microexon codes (depicted in Supplementary Table 5), suggesting that postsynaptic target cells might not instruct the identity of LAR-RPTP splice variants. Moreover, each LAR-RPTP member exhibits different expression repertoires of meA/meB microexons in different brain areas, cell types, and neural circuits. Intriguingly, we detected a previously unidentified microexon that is alternatively spliced in *Ptprd* and *Ptprf* (but not in *Ptprs*) and encodes 5 (15-bp) or 4 (12-bp) residues localized to the region between the eighth FN repeat of *Ptprd* and the transmembrane region of *Ptprf*. In addition, we found that the *Ptprs* meD microexon is not detected in mice, suggesting a possible evolutionary difference in alternative splicing programs for *Ptprs*[2], although the physiological role of the meD microexon in the brain is completely unknown.

The current study demonstrated that PTPδ actions in the context of specific hippocampal dCA1 circuits depend on the presence of an insert at meA, as recapitulated by electrophysiological and behavioral manifestations in both *Ptprd*-cKO and *Ptprd* meA-cKO mice, suggesting the possibility that PTPδ meA-mediated synaptic adhesion pathways are physiologically significant. Remarkably, we uncovered differential requirements for PTPδ in presynaptic neurons that project to hippocampal dCA1 pyramidal neurons, establishing previously unrecognized glutamatergic synaptic properties. Specifically, we found that PTPδ is required for NMDAR-mediated postsynaptic responses and asynchronous release in a subset of dCA1 circuits, indicating a non-canonical role of PTPδ as a presynaptic CAM. Moreover, the roles of PTPδ meA[+] variants in dictating different synaptic parameters manifested differently across distinct dCA1 circuits. To the best of our knowledge, these results are the unequivocal demonstration of the physiological significance of PTPδ meA splicing in organizing the diversity of distinct neural circuit architectures and mediating object-location memory. It remains to be determined whether IL-1RAcP and/or IL1RAPL1, the only known postsynaptic ligands that are specific to PTPδ[66,67], together with presynaptic PTPδ, are required to control the synaptic properties of the corresponding circuit. Notably, unlike PTPσ[42,68], presynaptic PTPδ is not required for the regulation of neurotransmitter release in the examined hippocampal dCA1 circuits. Collectively, our results deliver the important message that the expression level of a synaptic CAM and/or anatomical feature of the neural circuit alone cannot precisely predict its functional significance, underscoring the need to rigorously establish the circuit context-dependent roles of a synaptic CAM and microexons (in the case of LAR-RPTPs) in regulating specific synapse properties.

Expression of *Ptprd* meA, but not that of *Ptprd* meB, was altered in hippocampal DG neuronal engrams upon salient contextual fear conditioning. It is possible that activity-induced increased levels of *Ptprd*

meA[+] variants activate *trans*-interactions with IL1RAPL1 (or IL-1RAcP) and/or *cis*-interactions with Nrxns[15,66]. Because synaptic connections to DG engrams from the medial entorhinal cortex are critical for fear memory generalization[55], it will be worthwhile examining whether the interactions of PTPδ with known ligands through meA are distinctively engaged during various cognitive tasks. meA is divided into meA1 and meA2, which could further diversify combinatorial synaptic adhesion codes. Intriguingly, meA1 and meA2 appear to be differentially expressed in neurons and non-neurons, implying a distinct role for PTPδ meA-mediated adhesion codes in regulating neuron-microglia interactions. LAR-RPTPs are also expressed in various non-neuronal cells in the CNS[69–71]; thus, further studies are warranted to follow up on these intriguing observations and enhance our understanding of the diverse roles of LAR-RPTPs in the CNS.

Because *Ptprd* mRNAs are expressed primarily as meB[+] variants, the diversity of PTPδ-mediated synaptic adhesion pathways is likely dominated by the meA code. In contrast, *Ptprs* mRNAs are predominantly expressed as meA[−]meB[+] variants, indicating that the meB code is likely a major determinant underlying PTPσ-mediated *trans*-synaptic signaling involving Slitrks and SALMs[72,73]. The current study also revealed the dynamic expression patterns of *Ptprf* meA, highlighting the need to examine the synaptic function of LAR, which to date has been largely underexplored, relative to PTPδ and PTPσ. Moreover, how LAR-RPTP splice isoforms are selected for production in a specific type of cell and neural circuit should be determined by identifying RNA-binding proteins that regulate alternative splicing of LAR-RPTP mRNAs under basal and/or activity-dependent conditions. One prime candidate is nSR100 (neural-specific SR-related protein of 100 kDa), which promotes the inclusion of ~50% of the conserved neural microexons[19,22,74,75]. Intriguingly, nSR100 expression levels are closely correlated with the inclusion of neural microexons that are switched on late during neural differentiation[19]. Thus, future studies should systematically test whether the inclusion of meA and/or meB in all three LAR-RPTPs is subject to the nSR100-regulated splicing program and validate its physiological role in the context of specific neural circuits. Another intriguing avenue would be to explore the splicing machinery that operates specifically in astrocytes, which also express a significant amount of LAR-RPTP mRNA[76]. This could aid in testing the hypothesis that astrocytic LAR-RPTPs organize synaptic properties distinct from those regulated by neuronal LAR-RPTPs.

Given that neural microexons have been implicated in various neurological disorders, including autism spectrum disorders (ASDs), spinal muscular atrophy, myotonic dystrophy, frontotemporal dementia, and Lou Gehrig's disease[6,19], the current study will serve as a critical framework for dissecting neural circuits that are vulnerable to altered activities in these disorders. Production of transgenic knock-in mice in which microexon inclusion in all three LAR-RPTP mRNAs can be genetically manipulated would be beneficial for determining the contribution of dysregulation of specific microexons to shaping molecularly vulnerable neural circuits. While more work is required to understand the significance of synaptic adhesion codes signified by dynamic LAR-RPTP microexon profiles, the current study provides a sound platform for understanding how synaptic adhesion molecules shape the properties of various synapses and neural circuits.

## Limitations of the study

The current study has several caveats. First, we were unable to use an analysis of the interdependency among all splicing sites within single LAR-RPTP mRNA molecules to estimate the number of different LAR-RPTP splice variants or detect low-expressed LAR-RPTP isoforms because we did not use a long-read sequencing technology (e.g., PacBio single-molecule real-time sequencing, Oxford Nanopore sequencing). Second, for technical reasons, we were unable to fully probe the proteomic landscape of LAR-RPTP protein splice variants with or without a specific microexon-encoding peptide. Considered in light of

a recent computational analysis[77], it could be that one LAR-RPTP splice variant is translated more than others, possibly regulated by as yet unidentified microexon-dependent translational silencing machinery[78]. Third, the size of the FACS gate inherently serves to remove small cells, dendrites, and axonal remnants. Thus, the transcriptional signatures of LAR-RPTP microexons identified in the current study might not precisely reflect the expression patterns of the subset of LAR-RPTP mRNA subpopulations that are targeted to specific axonal terminals. Fourth, although we demonstrated differential functions of *Ptprd* meA inclusion in different neural circuits, the physiological significance of other LAR-RPTP microexons remains unclear. Moreover, despite enormous efforts, we were only able to electrophysiologically analyze three projections encompassing a single brain area (i.e., dCA1). Whether the phenotypes identified in the current study represent canonical functions of PTPδ remains to be determined. Because regulatory inclusion of PTPδ meA dictates the binding to IL1RAPL1, there is an urgent need to systematically investigate the involvement of IL1RAPL1 in distinct synaptic properties in the context of a defined neural circuit.

## Methods

### Mice
PTPσ floxed (*PTPσ[f/f]*), PTPδ floxed (*PTPδ[f/f]*), PTPδ meA floxed (*PTPδ meA[f/f]*) and PTPδ-tdTomato reporter mice were described previously[24,42,79]. The following reporter lines were purchased from The Jackson laboratory: Rosa26[LSL-tdTomato] (Ai9; Cat# 007909), *Sst*-IRES-Cre (Cat# 013044), *Emx1*-Cre (Cat# 005628), *Pvalb*-Cre (Cat# 008069), *Drd1*-Cre (Cat# 3836633), *Drd2*-Cre (Cat# 3836635), *Pvalb*-T2A-FlpO-D (Cat #022730), and *Sst*-IRES-FlpO (Cat# 028579). Only male mice were used for experiments (except those presented in Supplementary Fig. 3), and viral injections were performed in age-matched littermates.

### Animal husbandry and handling
All mice were maintained and handled in accordance with the animal care standards outlined in the Guide for the Care and Use of Experimental Animals and were approved by the Daegu Gyeongbuk Institute of Science and Technology (DGIST) Administrative Panel on Laboratory Animal Care (DGIST-IACUC-20102205-0003 and DGIST-IACUC-21060201-0010). Male adult mice on a C57BL/6 N background (purchased from Daehan Biolink) were used for all studies, except those presented in Supplementary Fig. 3. Mice were maintained at 24 °C on a 12:12-h light/dark cycle, with lights on at 7:00 and off at 19:00. Mice were given *ad libitum* access to food and water. Mice were weaned on postnatal day 28 (P28), and 2–5 mice were housed per cage to avoid social isolation and overcrowding.

### Antibodies
PTPδ peptides (IQKLTQIETGENVTGMELEF) and PTPσ peptides (PIADMAEHTERLKANDSLK) were synthesized and conjugated to keyhole limpet hemocyanin through a cysteine added to the C-terminus of the PTPδ peptide or the N-terminus of the PTPσ peptide. After immunization of rabbits with the respective immunogen, specific PTPδ antibodies (JK123; 1 μg/ml; RRID: AB_2713995) and PTPσ antibodies (JK125; 1 μg/ml; AB_2713993) were affinity-purified using a Sulfolink column (Pierce). The following primary antibodies were obtained commercially, as indicated: Goat polyclonal anti-GFP antibody (1:1000 dilution; RRID: AB_218182; Cat# 600-101-215) was purchased from Rockland Immunocytochemicals. FITC-AffiniPure donkey anti-goat IgG (1:150 dilution; RRID: AB_230401; Cat# 705-095-147) was purchased from Jackson ImmunoResearch.

### Plasmids
To generate pAAV-U6-mCherry, the mCherry sequence was PCR amplified from pAAV-Ef1a-DIO-DSE-mCherry-PSE-shRNA. AvrII-WPRE-

pA (purchased from Addgene; Cat #129669), the EGFP sequence was eliminated from pAAV-U6-EGFP by digestion with *Mlu*I and *BsrG*I, and the mCherry sequence was subcloned into the vector's *Mlu*I/*BsrG*I enzyme site. The following plasmids were obtained commercially: rAAV2-retro helper (Addgene; Cat# 81070), pAAV-RAM-d2TTA::TRE-NLS-mKate2-WPREpA (Addgene; Cat# 84474), pAAV-hSynI-FLEX-TVA-P2A-EGFP-2A-oG (Addgene; Cat# 85225), EnvA G-deleted Rabies-mCherry (Addgene; Cat# 32626), and pAAV-EF1α-fDIO-hM4Di-mCherry (Addgene; Cat# 50461). pAAV-EF1α-fDIO-NLS-GFP-ΔCre and pAAV-EF1α-fDIO-NLS-GFP-Cre were constructed by PCR amplification of the NLS-ΔCre-GFP and NLS-Cre-GFP segment from pAAV-hSynI-NLS-ΔCre-GFP and pAAV-hSynI-NLS-Cre-GFP vector, respectively, followed by subcloning into the pAAV-EF1α-fDIO at *BsrG*I and *Nhe*I sites. The following plasmids were previously described: pAAV-U6-EGFP[80]; pAAV-hSynI-ΔCre-EGFP and pAAV-hSynI-Cre-EGFP[42] and pAAV-phSynI-WGA-IRES-mCherry-Flpo-bGHGpA[81]; and pAAV-DIO-hM3Dq-2A-mCherry[82].

### RNA sequencing
7-week-old male mice were anesthetized by intraperitoneal injection of a saline-based 2% Avertin solution (2,2,2-tribromoethyl alcohol dissolved in tert-amyl alcohol [Sigma]) and then sacrificed for total RNA extractions. Extracted brain substructures were ground as finely as possible in liquid nitrogen, and 30 mg of tissue was lysed and used for extracting RNA with RNeasy Mini kit (Qiagen; Cat#74004). In all biological replicates, the following primer pairs for neuroanatomic marker genes were used to control for extraneous tissue cross-contamination: *Satb2*, 5′-CAG AGG TAC CAC GTG AAG CA3′ (forward) and 5′-GCC TGC GGA GTT CAC ATT AT-3′ (reverse); *Lef1*, 5′-CGG TTG TTT CGG AAA AAG AA-3′ (forward) and 5′-GGT CGC TGG CTT TCT AGT TG-3′ (reverse); *Prox1*, 5′-TTG CAA CGC TCT TTT GAA TG-3′ (forward) and 5′-CCC CTT GTG ATG AAG GAA AA-3′ (reverse); *Rgs9*, 5′-AGA AAT GCT GGC CAA AGC TA-3′ (forward) and 5′-GCA GCT CCT TTT TGA GTT GG-3′ (reverse); *Gabra6*, 5′-AGG AGT CAG TCC CAG CAA GA-3′ (forward) and 5′-GTT GAC AGC TGC GAA TTC AA-3′ (reverse). RNA integrity was assessed using an Agilent 2100 Bioanalyzer (Agilent). Quality-controlled RNA samples were used for RNA library preparation with a SureSelect[QXT] RNA Direct Library preparation kit (Agilent; Cat# G7564A) and target enrichment. At least 50 target enrichment probes were generated for each gene, achieving transcript targeting coverage greater than 98%. Target mRNAs used for library construction were captured by RNA hybridization probes printed onto microarrays. Libraries were prepared for 100-bp paired-end sequencing using a TruSeq RNA Sample Preparation Kit (Illumina; Cat# RS-122-2001) as described by the manufacturer. In brief, mRNA was purified from 2 μg of total RNA using oligo (dT) magnetic beads and fragmented. Single-stranded cDNA was synthesized from the resulting fragmented mRNAs by random hexamer priming and used as a template for preparing double-stranded cDNA. After sequential end repair, A-tailing, and adapter ligation processes, cDNA libraries were amplified by PCR. cDNA libraries were evaluated for quality using an Agilent 2100 BioAnalyzer and were quantified using a KAPA library quantification kit (Kapa Biosystems; Cat# KR0405) according to the manufacturer's library quantification protocol. Following cluster amplification of denatured templates, samples were sequenced as paired-end reads (2 × 100 bp) using Illumina HiSeq2500 (Illumina).

### RNA-seq data analysis
RNA-seq reads were aligned to the mm10 genome reference (GRCm38.92) using STAR (version 2.6.0a)[83] with the following parameters: outFilterMultimapNmax = 1; outSAMattributes = All; two-passMode = Basic. Genes annotated in mm10 were quantified using HTSeq-count[84]. To PSI values (denoted by Ψ) of LSVs in six different brain regions (cortex, olfactory bulb, thalamus, hippocampus,

striatum, and cerebellum) were quantified using STAR-aligned BAM files as input and analyzed with MAJIQ-PSI (v1.1.5) using default parameters. For the analyses of *Nrxn1* SS#4, we used BLAST[85] to search for the genomic coordinate of SS#4 (exon 22 in Supplementary Fig. 1d) based on the previously defined sequence, GGAAACAATGATAAACGAGCGCCTGGCGATTGCTAGACAGCGAATTCCATATCGACTTGGTCGAGTAGTTGATGAATGGCTACTCGACAAA, obtained from[86], identifying two LSVs containing microexon 31 (SS#4). The expected $\Psi$ ($E[\Psi]$) was obtained from the outputs of MAJIQ-Volia (v1.1.8) for these two LSVs across six brain regions. For LAR-RPTP mRNAs except for meA isoforms of *Ptprd*, we used MAJIQ-Volia to obtain $E[\Psi]$ values in the same way as was done for *Nrxn1*. However, for *Ptprd*, we quantified the expression of meA isoforms (meA1$^+$meA2$^+$, meA1$^+$meA2$^-$, meA1$^-$meA2$^+$, and meA1$^-$meA2$^-$) using kallisto (v0.46.2)[31]. The index was built with a k-mer parameter of 11, and expression levels were quantified using kallisto quant with default parameters. For the validation of other isoforms of *Ptprd*, *Ptprf*, and *Ptprs*, the k-mer parameter was set to 13 in most cases, for *Ptprf* meA isoforms, it was set to 23. The expression levels of these isoforms were then estimated using the default parameters of kallisto quant.

## Visualization of alternative splicing repertoires using *DEXSeq*

To investigate the alternative splicing repertoire of LAR-RPTP mRNAs, we employed the DEXSeq (v1.30.0) R package[30]. BAM files obtained from alignment using the STAR aligner were utilized, together with the GTF (GRCm38.92) file incorporating the microexon sequences of *Ptprs*, *Ptprd*, and *Ptprf*. Splicing patterns were visualized using the plotDEXSeq function with default option parameters.

## Polyacrylamide DNA electrophoresis for analysis of microexon dynamics

Semi-quantitative RT-PCR was performed using Q5 High-Fidelity DNA Polymerase (NEB; Cat# M0491L) as per the manufacturer's instructions using 100 ng of total RNA as a template in a 25 µL reaction. Reaction products were resolved by polyacrylamide gel electrophoresis (PAGE) on 10% gels, and bands were quantified using Image lab software (BIO-RAD). Primer sequences are provided in Supplementary Table 3. meA or meB microexons of *LAR-RPTP* mRNAs were quantified using the following primer pairs: *Ptprs* meA, 5′-GAG GTT GGC AGG TGA TGA GT-3′ (forward) and 5′-TAA GGA CTT CCT GCC TGT GG-3′ (reverse); *Ptprs* meB, 5′-TGA GTC CTT GAC ATC CGT GA-3′ (forward) and 5′-ACT CAT CAC CTG CCA ACC TC-3′ (reverse); *Ptprd* meA, 5′-TGG TGG CAA CAC ACT CGT AT-3′ (forward) and 5′-CGG ATC CAG AAA TCA CTT GG-3′ (reverse); *Ptprd* meB, 5′-TAC ACT CCA CCT GGC ATG A-3′ (forward) and 5′-CCG ACC AAG GAA AAT ACG AG-3′ (reverse); *Ptprf* meA, 5′-TCG CTG CTC TCT ATC TGC AA-3′ (forward) and 5′-CGA CTA TCG ACA TGG GAC CT-3′ (reverse); *Ptprf* meB, 5′-AGA GAG CAG CGA GGA GTC TG-3′ (forward) and 5′-CGA CTA TCG ACA TGG GAC CT-3′ (reverse); *Ptprd* e60, 5′-ACC CAC TAA TCA TGA AAT CA3′ (forward) and 5′-CGT CGA CAT AGC AAC A-3′ (reverse); and *Ptprd* e61, 5′-CAT CGA CTC ACA GGT TAG G-3′ (forward) and 5′-CTC TAT ACC CTT GGA TCT G-3′ (reverse). Q5 High-Fidelity DNA Polymerase (NEB) was used for PCR amplification. Amplified cDNAs were mixed with Midori Green Advanced DNA stain (NIPPON Genetics EUROPE GmbH; Cat# MG06) and separated by PAGE on 10% gels, which achieved quantifiable separation of microexons according to size. The patterns of *Nrxn1* splicing events at SS#4 sites were analyzed[87] using the primer pair, 5′-CTG GCC AGT TAT CGA ACG CT-3′ (forward) and 5′-GCG ATG TTG GCA TCG TTC TC-3′ (reverse). PCR amplicons were separated on 3% agarose gels using Midori Green advanced DNA stain (NIPPON Genetics EUROPE GmbH). DNA gel images were captured using a BIO-RAD gel doc imaging system, and band intensity ratio was quantified with Image Lab software (BIO-RAD) or ImageJ (NIH). Amplicons were extracted from the acrylamide gel for sequencing analyses using the crush and soak method[88].

## Confirmation of unidentified splicing events

Reliable cut-offs for the detection of novel exons in the MAJIQ RNA splicing pipeline were chosen by performing reverse transcription using primers specific for a designated novel exon or a novel exon flanking a canonical exon. The identified *Ptprd* exons were validated with the cortical cDNA library used for RNA-sequencing analyses and the following primers: e14-forward (5′-CTC GAA ATA GAA GGC AAG CA3′), e15-reverse (5′-CTG AGC ACT GGG TGC ATT T-3′), e16-reverse (5′-ACT AGG TTG TGG GCT GTT-3′), e25-forward (5′-CAT CGA CTC ACA GGT TAG G-3′), and e20-reverse (5′-GCT TGG ACT CTG GAT CGT-3′). Small amounts of splicing variants (> 10 reads) were detected by PCR using 100 ng of cDNA and PrimeSTAR polymerase (TaKaRa; Cat# R040A) in a total reaction volume of 25 µL, followed by extraction of PCR amplicons from the agarose gel and sequencing using the indicated forward primer of each PCR mixture.

## Quantitative reverse transcription polymerase chain reactions (qRT-PCRs)

RNA was quantified using NanoDrop One (Thermo Scientific), and 500 ng of total RNA was used to synthesize cDNA using ReverTra Ace-α-kit (Toyobo; Cat# FSK101). Quantitative PCR was performed with 100 ng of cDNA using CFX96 Touch Real-Time PCR (Bio-Rad). The ubiquitously expressed glyceraldehyde-3-phosphate dehydrogenase (*Gapdh*) gene was used as an endogenous control. The following primer pairs were used for PCR amplifications: *Satb2*, 5′-CAG AGG TAC CAC GTG AAG CA3′ (forward) and 5′- GCC TGC GGA GTT CAC ATT AT −3′ (reverse); *Lef1*, 5′- CGG TTG TTT CGG AAA AAG AA −3′ (forward) and 5′-GGT CGC TGG CTT TCT AGT TG-3′ (reverse); *Prox1*, 5′-TTG CAA CGC TCT TTT GAA TG-3′ (forward) and 5′-CCC CTT GTG ATG AAG GAA AA-3′ (reverse); *Rgs9*, 5′- AGA AAT GCT GGC CAA AGC TA −3′ (forward) and 5′-GCA GCT CCT TTT TGA GTT GG −3′ (reverse); *Gabra6*, 5′-AGG AGT CAG TCC CAG CAA GA-3′ (forward) and 5′-GTT GAC AGC TGC GAA TTC AA-3′ (reverse); and *Gapdh*, 5′-ACA TGG TCT ACA TGT TCC AG-3′ (forward) and 5′-TCG CTC CTG GAA GAT GGT GAT-3′ (reverse).

## Shotgun mass spectrometry

1. Preparation of materials. Iodoacetamide (IAA), 1,4-dithiothreitol (DTT), formic acid (FA), and ammonium bicarbonate (ABC) were purchased from Sigma-Aldrich. Trypsin was purchased from Promega (Madison, WI, USA). Acetonitrile (ACN; HPLC grade) and high-grade water were purchased from Merck (Darmstadt, Germany). Heavy, stable isotope-labeled peptides with C-terminal [U-13C6, 15N2] lysine or [U-13C6, 15N4] arginine for all target peptides (> 95% purity) were chemically synthesized by ANYGEN Co. (Gwangju, Korea).

2. Preparation of brain samples. Hippocampal and cortical homogenates (1 mg) were obtained from 7-week-old male mice. The indicated lysates were subjected to immunoprecipitation with anti-PTPδ antibodies (JK123) overnight at 4 °C, after which 30 µL of a 1:1 suspension of protein A-Sepharose (GE Healthcare) was added, and the mixture was incubated for 2 h at 4 °C with gentle agitation. Immuno-complexes were then resolved by SDS-PAGE on 10% acrylamide gels, after which the bands in gels were sliced, detained with 30% MeOH and 50% ACN 10 mM ABC buffer, and digested as follows. Briefly, sliced bands were reduced by incubating with 10 mM DTT (in 100 mM ABC) at 56 °C for 1 h and alkylated by incubating with 55 mM IAA (in 100 mM ABC) in the dark at room temperature for 45 min. Each band was washed with deionized water and 100% ACN, swollen in digestion buffer containing 50 mM ABC, 5 mM CaCl₂, and 1 µg trypsin; and incubated at 37 °C for 12−16 h. After quenching the digestion using 50% ACN containing 0.1% FA, peptides were recovered by extracting once with 100% ACN and twice with 50% ACN containing 0.1% formic acid (FA).

The resulting peptide extracts for each band were pooled, lyophilized, and stored at −20 °C until use.

3. Bottom-up LC–MS/MS. Dissolved samples in mobile phase A (99.9% water containing 0.1% FA) were analyzed using an LC–MS/MS system consisting of an UltiMate 3000 RSLCnano system and an Orbitrap Fusion Lumos mass spectrometer (Thermo Scientific) equipped with a nanoelectrospray source. An autosampler was used to load the sample solutions into a $C_{18}$ trap column (Acclaim™ PepMap™ 100; NanoViper; 3 μm particle, 75 μm × 2 cm; Thermo Fisher Scientific). The samples were trapped, desalted, and concentrated on the trap column for 8 min at a flow rate of 3 μL/min, and then separated on a $C_{18}$ analytical column (PepMap™ RSLC $C_{18}$, 2 μm, 100 Å, 75 μm × 50 cm; Thermo Fisher Scientific). The mobile phases were composed of 99.9% water (A) and 99.9% ACN (B), and each contained 0.1% FA. The flow rate was set at 250 nL/min. The LC gradient started at 1% of B for 10 min and was ramped to 5% B for 5 min, 18% B for 70 min, 30% B for 15 min, and 95% B for 1 min, and then was held at 95% B for 8 min and 95% B for an additional 1 min. The column was re-equilibrated with 5% B for 10 min before the next run. Ions were produced by applying a voltage of 1800 V. During the chromatographic separation, the Orbitrap Fusion Lumos mass spectrometer (Thermo Fisher Scientific) was operated in data-dependent mode, automatically switching between MS1 and MS2. Full-scan MS1 spectra (400–2000 m/z or 300–1200 m/z) were acquired by the Orbitrap, with a maximum ion injection time of 50 ms at a resolution of 120,000 and an automatic gain control (AGC) target value of $4.0 × 10^5$. The MS2 spectra were acquired with an Orbitrap mass analyzer at a resolution of 30,000 with HCD fragmentation (normalized collision energy, 27%; maximum ion injection time, 54 ms, AGC target value, $5.0 × 10^4$). Previously fragmented ions were excluded for 30 s within 10 ppm. Internal calibration was conducted with the mass peak set at 445.12003 m/z, which releases polysiloxane from the silica capillary of the NanoSprayer[89].

4. Identification of PTPδ protein. MS/MS spectra were analyzed using the IP2 search algorithm (Integrated Proteomics Applications, Inc., San Diego) with the PTPRD protein sequence (release date, October 2022) containing the contaminant proteins. The reversed sequences of all proteins were added to the database, and false discovery rates (FDRs) were calculated. ProLucid[90] from the integrated proteomics pipeline (IP2; Integrated Proteomics Applications, Inc., San Diego, CA, USA) was used to identify the peptides, with a precursor mass error of 5 ppm and fragment ion mass error of 50 ppm. Trypsin was selected as the enzyme, with three potential missed cleavages and partial tryptic cleavage allowed. Carbamidomethylation at cysteine and oxidation at methionine were chosen as static and variable modifications, respectively. The output data files were used to devise a protein list using DTASelect software[91] (The Scripps Research Institute, San Diego, CA, USA), with an FDR < 0.1.

## Nano liquid chromatography-parallel reaction monitoring (LC-PRM)

1. Selection of peptides. The following peptides were selected based on previously reported information obtained for meA peptide and/or peptide sequences detected by shotgun LC–MS/MS analyses: SGALQIEQSEESDQGK (for meA1⁻A2⁻); SESIGGTPIR (for meA1⁺A2⁺); SGGTPIR (for meA1⁻A2⁺); and SESIGALQIEQSEESDQGK (for meA1⁺A2⁻). The peptides NVLELNDVR and VVAVNNIGR encoding *Ptprd* exon 60 or exon 61, respectively, were used as controls.

2. Optimization of PRM assays for target peptides. Six heavy peptide mixtures at 50 fmol/μL for each peptide except SGGTPIR

(50 pmol/μL) were created by mixing in the stock solution of heavy peptide (1 nmol/μL in deionized water containing 3% FA). Synthesized heavy peptide quality and fragment ions were verified, and retention time (from LC separation) and best charge state for each peptide were monitored, by performing LC–MS/MS analysis with higher-energy collisional dissociation on an Orbitrap Exploris 480 mass spectrometer equipped with an UltiMate 3000 RSLCnano system (Thermo Scientific).

3. LC-PRM analyses. Heavy internal standard mixtures–100 fmol/μL for each peptide except for SGGTPIR (100 pmol/μL)–were prepared and equally spiked into digested homogenates from the cortex, hippocampus, or striatum of 7-week-old male mice at a concentration of 50 fmol/μL (except for SGGTPIR, which was spiked at 50 pmol/μL). Prepared samples in mobile phase A (99.9% water containing 0.1% FA) were analyzed using an LC–MS/MS system consisting of an UltiMate 3000 RSLCnano system and an Orbitrap Exploris 480 mass spectrometer (Thermo Scientific) equipped with a nanoelectrospray source. An autosampler was used to load the sample solutions onto a $C_{18}$ cartridge column (PepMap-neo $C_{18}$, 5 μm particle, 300 μm × 5 mm; Thermo Fisher Scientific). Samples were trapped and then desalted and concentrated on the cartridge column for 8 min at a flow rate of 20 μL/min; trapped samples were then separated on a $C_{18}$ analytical column (PepMap-RSLC $C_{18}$, 2 μm, 100 Å, 75 μm × 50 cm; Thermo Fisher Scientific). The mobile phases consisted of 99.9% water (A) and 99.9% ACN (B), and each contained 0.1% FA. The flow rate was set at 250 nL/min. The LC gradient started with 5% B for 9 min and was ramped to 25% B for 56 min, 35% B for 5 min, and 95% B for 1 min, and then held at 95% B for 9 min and 5% B for an additional 1 min. The column was re-equilibrated with 5% B for 9 min before the next run. Ions were produced by applying a voltage of 1800 V. During the chromatographic separation, the Orbitrap Exploris 480 mass spectrometer (Thermo Fisher Scientific) was operated in targeted MS/MS mode for PRM analysis, automatically switching between MS1 and targeted MS2. Full-scan MS1 spectra (300–1500 m/z) were acquired by the Orbitrap, with a maximum ion injection time of 100 ms at a resolution of 120,000 and an AGC target value of $3.0 × 10^6$ (300%). Targeted MS2 spectra were acquired by the Orbitrap mass analyzer at a resolution of 45,000 with HCD fragmentation (normalized collision energy, 30%; maximum ion injection time, 120 ms; AGC target value, $2.0 × 10^5$ [200%]) within a 0.7-Da window. The m/z value of endogenous target peptides and their corresponding heavy peptides are listed in Supplementary Table 4 for LC-PRM analyses.

4. Data processing. The resulting raw data were analyzed using Skyline daily 22.2.1.425[92]. The quality of endogenous peptides data was verified by comparing the peak shape and retention time stability of fragment ions of endogenous peptides with those of heavy peptides. For quantitative analyses, the three most abundant fragment ions from each peptide were selected based on those of the heavy peptide. The sum intensity of the three fragment ions was used in calculating the ratio of endogenous to heavy peptide (L/H ratio). Finally, to account for batch-to-batch variations in PTPδ protein enrichment by immunoprecipitation and differences in abundance induced in different mouse brain tissue samples, we normalized the L/H ratio of each peptide by dividing it by the L/H ratio of a one- or two-peptide references.

## Preparation of recombinant AAVs

HEK293T cells were co-transfected with the indicated AAV vectors and pHelper and pAAV1.0 (serotype 2/9) vectors. After culturing for 72 h, transfected HEK293T cells were collected, lysed, mixed with 40% polyethylene glycol and 2.5 M NaCl, and incubated for 1 h at 4 °C, then centrifuged at $2000 × g$ for 30 min. The pellets were resuspended in

HEPES buffer (20 mM HEPES, 115 mM NaCl, 1.2 mM CaCl$_2$, 1.2 mM MgCl$_2$, 2.4 mM KH$_2$PO$_4$) and an equal volume of chloroform was added. The mixture was centrifuged at $400 \times g$ for 5 min and concentrated three times using a Centriprep centrifugal filter (Millipore) at $2000 \times g$ for 30 min and once using an Amicon Ultra centrifugal filter (Millipore) at $16,000 \times g$ for 30 min. Before titration of AAVs, contaminating plasmid DNA was eliminated by treating 1 μL of concentrated, sterile-filtered AAVs with 1 μL of DNase I (Sigma-Aldrich) for 30 min at 37 °C. After treatment with 1 μL of stop solution (50 mM ethylenediaminetetraacetic acid) for 10 min at 65 °C, 10 μg of proteinase K (Sigma-Aldrich) was added and AAVs were incubated for 1 h at 50 °C. Reactions were inactivated by incubating samples for 20 min at 95 °C. The final virus titer was quantified by qPCR detection of EGFP or mCherry sequences and subsequent reference to a standard curve generated using pAAV-U6-GFP or pAAV-T2A-mCherry, respectively. All plasmids were purified using a Plasmid Midi Kit (Qiagen).

### Preparation of rabies viruses
EnvA-pseudotyped, G-deleted rabies viruses were produced using EnvA G-deleted rabies-mCherry[93]. Packaged pseudotyped delta G rabies virus (Salk; titers of $> 5 \times 10^7$ infectious units per milliliter) was used for cell-type–specific, monosynaptic, retrograde labeling.

### Stereotaxic surgery and virus injection
5-week-old male mice were anesthetized by intraperitoneal injection of a saline-based 2% Avertin solution (2,2,2-tribromoethyl alcohol dissolved in tert-amyl alcohol [Sigma]), and their heads were fixed in a stereotactic apparatus. Recombinant AAV viruses were injected into the hippocampal CA1 region (coordinates: AP −2.1 mm, ML ± 1.3 mm, and DV −1.8 mm), mPFC (coordinates: AP + 1.8 mm, ML ± 0.4 mm, and DV −2.3 mm), or hippocampal DG (coordinates: AP −2.1 mm, ML ± 1.3 mm, and DV 2.3 mm) with a Nanofil syringe at a flow rate of 100 nl/min (injected volume, 300 nl) using a Nanoliter 2010 Injector (World Precision Instruments). Each injected mouse was restored to its home cage for 3–4 wk and used subsequently for FACS, confocal microscope imaging, electrophysiological recordings, or behavioral analyses.

### Cell-type–specific RNA preparation
Neurons were labeled in a fluorescent cell-type–specific manner by crossing *Sst*-IRES-Cre, *Emx1*-Cre, *Pvalb*-Cre, *Drd1*-Cre, and *Drd2*-Cre lines with Rosa26$^{LSL-tdTomato}$. All experiments were performed using double heterozygotic 7-week-old male mice. Mouse brains were immersed in FSC22 Frozen Section Media (Cat# 3801480; Leica Biosystems) and frozen at −80 °C. Frozen tissues were placed at −35 °C for 2 h before micro-sectioning and sliced into 40-μm–thick coronal sections using a cryotome (Model CM-3050-S; Leica Biosystems). Fluorescent labeling was detected using Axio Scan.Z1 (Zeiss). A reliable number of cells for FACS-mediated cell sorting was obtained by pooling brain tissue from at least three different mice and dissociating it into individual cells.

### Neural projection labeling using retrograde AAVs and rabies viruses
dCA1- or mPFC-projecting neuronal populations were labeled by injecting retrograde AAV-U6-GFP or AAV-U6-mCherry viruses into the dCA1 or mPFC, respectively, of 5-week-old male mice. Somatostatin- or parvalbumin-positive interneurons projecting to neurons in the CA3, SuB, or EC were labeled by injecting 5-week-old male *Sst*-ires-Cre and *Pvalb*-Cre mice in the CA1 with pAAV-hSynI-FLEX-TVA or EnvA G-deleted rabies-mCherry, respectively. After surgery, injected mice were kept in their home cage for 3 weeks to enable infected viral expression and fluorescence. Mouse brain slices were prepared as described in "Cell-type–specific RNA preparation".

### Labeling of fear memory neuronal ensembles using AAV-Fos–dependent RAM
Fear memory-associated, Fos-dependent neuronal ensemble populations were labeled by injecting AAVs expressing *F*-RAM into the DG of 5-week-old male mouse brains. After stereotaxic injection, 1 mg/ml doxycycline (Sigma-Aldrich; Cat# D9891) was administered for 7 days to prevent irrelevant engram labeling. Doxycycline was withdrawn 1 day before contextual fear conditioning to allow *F*-RAM-specific fluorescence labeling. After contextual fear conditioning and contextual fear retrieval, mice were sacrificed for immunohistochemistry or FACS.

### Fluorescence-activated cell sorting (FACS)
Mice were anesthetized by intraperitoneal injection of a saline-based 2% Avertin solution (2,2,2-tribromoethyl alcohol dissolved in tert-amyl alcohol [Sigma]). A papain dissociation system (Worthington; Cat# LK003150) was used to dissociate mouse brains. Dissected brain tissues were minced with a razor blade and incubated in a Sterile Earle's Balanced Salt Solution (SEBSS) containing papain (20 units/μL), DNase (2,000 units/μL), and actinomycin-D (5 μg/ml; Sigma; Cat# 50-76-0) for 1.5 h at 37 °C with gentle shaking. Debris and broken cells were subsequently removed by gradient centrifugation, after which cell pellets were suspended in cold EBSS containing actinomycin-D and subjected to FACS analyses using FACS Aria III (BD) with a 100 μm nozzle. Fluorescence-negative control littermate brain tissues were dissociated in parallel for use as negative controls. To avoid cell doublets and non-cell particles, we set the cell size gate around the forward scatter level, and sorted cells using the four-way purity method into 1.5 ml microcentrifuge tubes. Distinct fluorescence-positive neuronal populations ranging from $10^5$ to $10^9$ cells were sorted until 200,000-300,000 events were acquired. The sorted samples were centrifuged at $300 \times g$ for 7 min, after which supernatants were removed and pellets were stored at −80 °C until further use.

### Retrograde tracing
5-week-old male mice were anesthetized by intraperitoneal injection of a saline-based 2% Avertin solution (2,2,2-tribromoethyl alcohol dissolved in tert-amyl alcohol [Sigma]) and injected in the dCA1 area (AP: −2.0, ML: ±1.6, DV: −1.5) with rAAV$_2$-retro and 4 weeks later were perfused with 4% paraformaldehyde (PFA). Injected mouse brains were then removed, fixed overnight at 4 °C, and placed in phosphate-buffered saline (PBS) containing 30% sucrose until the tissue sank. Whole brains were sliced into sagittal sections (40 μm thickness) using a cryotome (Model CM-3050-S; Leica Biosystems); sections were then incubated overnight at 4 °C with anti-EGFP antibody (Rockland; Cat# 600-101-215; 1:1000) in blocking solution consisting of PBS containing 5% normal donkey serum albumin, 1 mg/ml bovine serum albumen and 0.5% Triton X-100. Sections were subsequently washed twice with PBS (5 min each) and then incubated with FITC-conjugated anti-goat secondary antibodies. Whole selected brain sections were scanned using a slide scanner (Axio Scan.Z1; Carl Zeiss) with a 20× objective lens, with all image settings kept constant during image acquisition. Z-stack images obtained with the slide scanner were converted to maximal projections, and the acquired images were further processed using ZEN software installed in Axio Scan.Z1. Higher resolution Z-stack images of CA3, SuB, and EC were obtained separately by confocal microscopy (LSM780; Carl Zeiss) with a 20× object lens.

### Chemogenetic manipulation with DREADDs
Male WT mice (5–6 weeks old) were anesthetized and injected with 200 nL of WGA-Flpo in CA1 and 300 nL of fDIO-Cre+DIO-hM4Di(Gi)-mCherry or DIO-hM3D(Gq)-mCherry in SuB, as described in "Stereotaxic surgery and virus injection". The mice were allowed to recover for 3 weeks, and then a solution containing deschloroclozapine (DCZ; HY-42110, MedChemExpress) or vehicle (saline) was intraperitoneally

injected at 24 h prior to the behavioral or electrophysiological experiments. DCZ was dissolved in 1–2% dimethyl sulfoxide (DMSO) in saline to a final concentration of 100 µg per kg.

## Electrophysiology

Transverse hippocampal slices (300 µm thick) were prepared from 8–10-week-old male mice. After anesthetizing with isoflurane, mice were decapitated and their brains were rapidly removed and placed in an ice-cold, oxygenated (95% $O_2$ and 5% $CO_2$), low-$Ca^{2+}$/high-$Mg^{2+}$ solution containing 3.3 mM KCl, 1.3 mM $NaH_2PO_4$, 26 mM $NaHCO_3$, 11 mM D-glucose, 0.5 mM $CaCl_2$, 10 mM $MgCl_2$, and 211 mM sucrose. Hippocampal slices were cut with a vibratome (VT1000s; Leica) and transferred to a holding chamber containing oxygenated artificial cerebrospinal fluid (aCSF) consisting of 124 mM NaCl, 3.3 mM KCl, 1.3 mM $NaH_2PO_4$, 26 mM $NaHCO_3$, 11 mM D-glucose, 2 mM $CaCl_2$, and 1 mM $MgCl_2$. Slices, incubated at 30 °C for at least 60 min before use, were placed in the recording chamber and perfused continuously with normal aCSF bubbled with 95% $O_2$ and 5% $CO_2$. All experiments were performed at 29–32 °C, and slices were used within 4 h. For measurement of evoked and asynchronous EPSCs (eEPSCs and aEPSCs, respectively), patch pipettes were filled with an internal solution consisting of 130 mM Cs-methanesulfonate, 5 mM TEA-Cl, 8 mM NaCl, 0.5 mM EGTA, 10 mM HEPES, 4 mM Mg-ATP, 0.4 mM Na-GTP, 1 mM QX-314, and 10 mM disodium phosphocreatine, adjusted to pH 7.3 with CsOH. The osmolarity of the intracellular solution was 280–290 mOsm. Electrical stimulation was applied using a concentric bipolar electrode (FHC), placed in the *stratum oriens*, *stratum radiatum*, or *stratum lacunosum moleculare*. eEPSCs were recorded at −70 mV (for AMPA-EPSCs) and +40 mV (for NMDA-EPSCs; 50 ms after stimulation). For eEPSC and aEPSC recordings, picrotoxin (50 µM; Tocris) was included to block $GABA_A$ receptors. aEPSCs were recorded in aCSF consisting of 124 mM NaCl, 3.3 mM KCl, 1.3 mM $NaH_2PO_4$, 26 mM $NaHCO_3$, 11 mM D-glucose, 8 mM $SrCl_2$, and 3 mM $MgCl_2$. The asynchronous release was induced by 10 Hz paired stimulation and measured for 5 s after the second pulse for 10 consecutive sweeps. To assess the intrinsic excitability of subicular pyramidal neurons, glass pipettes were filled with an intracellular solution that consisted of 130 mM K-gluconate, 20 mM KCl, 0.2 mM EGTA, 10 mM HEPES, 4 mM Mg-ATP, 0.3 mM Na-GTP, and 10 mM disodium phosphocreatine, and was adjusted to pH 7.3 with KOH. The injected current ranged from 0 pA to 300 pA at 25-pA increments. DCZ (10 µM; MedChemExpress) was applied to the hippocampal slices.

## Behavioral analyses

1. Open-field test. 9-week-old male mice were allowed to freely explore the environment for 10 min in a black acrylic open-field box (40 × 40 × 40 cm) in dim light (< 20 lux). The traveled distance and time spent in the center zone were recorded by a top-view camera and analyzed using EthoVision XT 10.5 software (Noldus).

2. Novel object-recognition test. An open-field chamber was used in this test. For training sessions, two identical objects were placed in the center of the chamber at regular intervals, and mice were allowed to explore the objects for 10 min. After the training session, mice were returned to their home cage for 24 h. For novel object-recognition tests, one of the two objects was randomly replaced with a new object, placed in the same position in the chamber. Mice were returned to the chamber and allowed to explore freely for 10 min. The number and duration of contacts with objects were analyzed using EthoVision XT 10.5 (Noldus) and used to determine a discrimination index, calculated as the difference between time spent exploring novel and familiar objects during the test phase.

3. Contextual fear-conditioning test. A fear-conditioning chamber (300 × 200 × 200 mm; Coulbourn Inc.) was used in this test. Mice were transferred to the conditioning chamber and allowed to move freely for 120 s, after which three 20-s auditory cues (conditioned stimulus [CS]) were sounded; a 0.55-mA footshock (unconditioned stimulus [US]) was administered during the last 2 s of the sound to induce pairing of CS and US. After conditioning, the mice were returned to their home cage. For the contextual fear test, mice were placed in the conditioning chamber 24 h later and allowed to move freely for 300 s in the absence of a stimulus. On the next day, the mice were placed in a different chamber scented with vanilla to provide a new context for 300 s. After 4 h, mice were allowed to explore the pre-exposed conditioning chamber for 300 s, with CS provided during the last 180 s. The freezing behavior of subjects was analyzed using FreezeFrame 5 software (Coulbourn Inc.) and presented as a percentage of the time.

4. Object-location memory test. All mice were habituated to the open-field area (40 × 40 × 40 cm) in dim light (< 20 lux) for 6 min per day for three consecutive days without objects. After habituation, two identical objects were placed in the open-field box and the mouse was allowed to freely explore the open field for 10 min. Twenty-four hours later, one object location was moved, and the mouse was allowed to freely explore the open field. The traveled distance and time spent on each object were recorded by a top-view camera and analyzed using EthoVision XT 10.5 software (Noldus).

5. Acoustic startle prepulse inhibition test. On the first day, the session was preceded by a 5-min exposure to 65-dB background noise to test acoustic startle responses. Each mouse then received 44 pulse trials consisting of no-stimulation, 75, 80, 85, 85, 90, 95, 100, 105, 110, 115, and 120 dB; this stimulation was semi-randomly repeated four times. Twenty-four hours later, the session was preceded by a 5-min exposure to 65-dB background noise to monitor prepulse inhibition, after which each mouse received 60 trials with 10–20 inter-trial intervals consisting of no-stimulation, 73, 76, 79, and 82 dB, again semi-randomly repeated four times. The percent prepulse inhibition was calculated as mean prepulse response/mean pulse response × 100. The startle at each pulse level was averaged across trials.

6. Three-chamber test. The testing apparatus consisted of a white acrylic box divided into three chambers (each 20 × 40 × 22 cm) with small holes on the dividing walls. Wire cups were placed at the edges of both side chambers. Testing mouse was placed in the chamber for 10 min to habituated to the environment. After then, an age-matched stranger mouse was placed in one of the wire cup and the sociability of the subject was assessed by measuring subjects' exploration times for the enclosed stranger mouse and the empty cup for 10 min. In the last session, a new stranger mouse was placed into rest of the empty wire cup, and social novelty was assessed by measuring subjects' exploration times.

## Statistical analyses

Data analyses and statistical tests were performed using GraphPad Prism 8.0 software (RRID: SCR_002798). All data are expressed as means ± standard errors unless stated otherwise. All experiments were performed using at least three independent mice, cultures, and/or cohorts of grouped mice. No statistical methods were used to pre-determine sample size and experiments were not randomized. Data were assessed with Student's *t*-test or one-way analysis of variance (ANOVA) using a non-parametric Kruskal–Wallis test, followed by Dunn's multiple comparison test for *post hoc* group comparisons, paired *t*-test or Mann–Whitney *U* test; '*n*' numbers used are presented in figure legends. Numbers shown indicate replicates and tests used to determine statistical significance are stated in the text and legends of

figures depict the results of the respective experiments. A *p*-value < 0.05 was considered statistically significant, and individual *p*-values are indicated in the respective figure legend.

## Reporting summary

Further information on research design is available in the Nature Portfolio Reporting Summary linked to this article.

## Data availability

The raw RNA-seq data generated in this study have been deposited in the NCBI BioProject under accession code PRJNA943606 (https://dataview.ncbi.nlm.nih.gov/object/PRJNA943606?reviewer=qb6tg4pvrhm5gr7rk468b5l63l). The proteome data generated in this study have been deposited in the MassIVE database under accession code MSV000091879 (https://massive.ucsd.edu/ProteoSAFe/dataset.jsp?task=1e036c7781474927b9e5593d32feb7f9). A reporting summary for this article is available as a Supplementary Information file. The main data supporting the findings of this study are available within the article and its Supplementary Figs. The source data underlying Figs. 1–9, Supplementary Fig. 1–3, Supplementary Fig. 5 and Supplementary Figs. 7–9 is provided as a Source Data file. Additional details on datasets and protocols that support the findings of this study will be made available by the corresponding author upon request. Source data are provided with this paper The datasets presented in this study are included in full wherever possible, and source data are provided within this paper.

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

## Acknowledgements

We are grateful to Jinha Kim (DGIST, Korea) for technical assistance, Drs. Eunjoon Kim (KAIST/IBS, Korea) and Hee-Sup Shin (IBS, Korea) for providing various mouse lines. This work was supported by the Korea Healthcare Technology R & D Project, funded by the Ministry for Health and Welfare Affairs, Korea (HI17C00080 to J.Ko), the DGIST R&D Program of the Ministry of Science and ICT (23-CoE-BT-01 to J.W.U. and J.Ko), the National Creative Research Initiative Program of the Ministry of Science and ICT (2022R1A3B1077206 to J.Ko), and the National Research Foundation of Korea (NRF) funded by the Ministry of Science and ICT (2021R1C1C2010767 to K.A.H.; 2021R1F1A1061840 to J.Y.L.; 2017M3C7A1048448 to J.K.K.; RS-2023-00207834 and 2023R1A2C2002535 to J.W.U.).

## Author contributions

K.A.H., T.H.Y. and J.Ko conceived the project. T.H.Y. performed the RNA-seq analyses with assistance from J.L. and J.K.K. K.A.H. and T.H.Y. performed immunohistochemical analyses. J.Ki carried out electrophysiological analyses. K.A.H. and J.Y.L. performed targeted mass spectroscopic analyses. K.A.H. and T.H.Y. managed the mouse breeding with assistance from G.J. K.A.H. performed mouse behavioral analyses. J.W.U., J.K.K. and J.Ko supervised the project. J.Ko drafted the manuscript with input from K.A.H., T.H.Y., J.Ki, J.Y.L. and J.K.K. All authors reviewed and edited the final manuscript.

## Competing interests

The authors declare no competing interests.
