## [Peer Review File · Nature Communications]

Specification of neural circuit architecture shaped by context-dependent patterned LAR-RPTP microexonsREVIEWER COMMENTS

Reviewer #1 (Remarks to the Author):

This study from the Ko lab focuses on profiling the spatial regulation of LAR-RPTP microexons in the mouse brain. Targeted RNA-Sequencing of various dissected brain regions as well as RT-PCR validations allowed the quantifications of microexon-A (meA) and microexon-B (meB) across the various LAR-RPTP paralogs, an analysis that didn't result in the observation of striking differences in the inclusion levels of individual microexons across the examined brain regions. The authors further complement this profiling by applying proteomics in an attempt to quantify microexon peptide levels. To investigate if there is cell (sub)type specific LAR-RPTP microexon regulation the authors applied genetic models to sort specific cell types and profiled meA and meB inclusion levels across various brain regions. More interestingly the authors show that Ptpd and Ptpf meA inclusion is promoted by experience-triggered neuronal signaling. The final part of the manuscript takes advantage of Ptpd gene and meA conditional knock-out mouse models that have been recently developed by the Eunjoon Kim laboratory and describes detailed electrophysiological recordings as well as behavioral tests that reveal deficits in object-location memory.

Although this study provides some interesting new insights of the in vivo role of Ptpd meA it is rather descriptive and lacks follow-up mechanistic investigation. For example, the authors show an increase of Ptpd meA inclusion in fear-memory activated hippocampal neurons as well as splicing differences in neuronal sub-types, but they fail to address which are the (splicing) factors responsible for this regulation, how do they respond to activity or how their expression changes in different cell types. Furthermore, the novel information provided could be considered as incremental given the previous work from the Kim lab (Park et al., EMBO, 2020). Although this is ultimately an editorial decision, this reviewer thinks that the study would be more suitable for a specialized journal.

Specific (minor) points:

1. The introduction could be improved by referring to the recent findings regarding the in vivo role of Ptpd meA (Park et al.,) as well as to previous studies characterizing the molecular function of these microexons (e.g. Yoshida et al., 2012).
2. The authors should provide both kallisto and MAJIQ data for all exons analyzed instead of cherry picking.
3. "Ptpd meA+ variants were prominently detected in the olfactory bulb.". This is an overstatement as based on the provided data meA is barely included in Ptpd.
4. The RT-PCR data provided in Figure 1f, in contrast to their quantification and RNA-Seq data, indicate highest inclusion of Ptpd meB in striatum (~50%). I assume this is not a representative picture.
5. A right bracket should be included after Drd2-Cre (line 7, page 9).
6. The structure of the last 2 figures doesn't agree with the text. The text is split into electrophysiological recording and behavioral tests while the figures are split into Ptpd gene or microexon KO mouse models.
7. Cell adhesion molecules should be spelt out in the first line of the discussion.

Reviewer #2 (Remarks to the Author):

Han et al. developed a novel method to precisely detect meA and meB expression and found cell-type specific patterns of meA and meB in Ptpd mRNAs. Alternative splicing of Ptpd occurred in an activity-dependent manner. Moreover, conditional depletion of Ptpd or Ptpd meA in the SuB/CA1 circuit strikingly increased excitatory synaptic transmission through NMDAR and further decreased objection-location memory. As the biological significance of alternative splicing in neural circuit has been extensively studied and proved for Nr1n1, the current study does not necessarily present a novel

concept. Nevertheless, the authors for the first time addressed and demonstrated the significance of microexon segments, particularly meA and meB of the LAR-RPTP family, in neural circuit specification.

Major points

1. The authors claimed that LAR-RPTPs utilize microexons to shape combinatorial synaptic adhesion pathways in distinct neural circuits. Indeed, they showed alterations of NMDAR-mediated synaptic transmission and objection-location memory in conditional depletion of Ptprd meA in the SuB/CA1 circuit. However, it remains elusive whether the neural connection between SuB and CA1 was altered.
2. The authors showed an increased Ptprd meA inclusion in fear memory engrams of the dentate gyrus (Fig. 5). This activity-dependent alternative splicing was not linked to other findings of synaptic activity or behavior altered by Ptprd meA depletion (Fig. 6, 7). In other words, findings shown in Fig. 5 and Fig. 6,7 are independent stories, although each finding is important. If the activity-dependent alternative splicing and the microexon-dependent synaptic activity and behavior are linked, it would strengthen the study.

Minor points

1. Genotype changes of Ptprd in SuB induced by WGA-Flpo and AAV-fDIO-Cre should be confirmed in Fig. 6 and 7.

Reviewer #3 (Remarks to the Author):

This research utilized intensive transcriptome and proteomics analyses to investigate the insertion patterns of the microexons, meA and meB, in LAR-RPTP family proteins. This was done across different brain regions, cell types, and neural projections. The study further delved into the correlation between these patterns and their effects on neural circuit functionality and in vivo behavior.

1. The researchers employed targeted RNA-seq with high sequencing read depth to decipher the spatiotemporal expression profiles and differential splicing of two microexons—meA and meB—of LAR-RPTP mRNAs.
2. The results from targeted RNA deep sequencing were authenticated using techniques like polyacrylamide gel electrophoresis, quantitative PCR, and Sanger sequencing.
3. PRM mass spectrometry quantification was used to examine PTP δ splice variants containing or lacking meA. It was observed that the abundance of each PTP δ mRNA splice isoform aligns with the corresponding PTP δ proteoforms in three distinct brain areas.
4. The investigation revealed that the meA/meB microexons of LAR-RPTPs exhibit varied expression patterns, even when observed within identical cell types in different brain areas. It was noted that neurons with common long-range projection targets or specific cell connectivity do not consistently use the same microexon codes.
5. The study further showcased that the functions of PTP δ in hippocampal dCA1 circuits are influenced by the presence of an insert at meA. This presence significantly impacts electrophysiological and behavioral attributes in specific mouse models.
6. PTP δ is essential for NMDAR-mediated postsynaptic responses and asynchronous release in a subset of dCA1 circuits.
7. Changes were observed in the expression of Ptprd meA, but not Ptprd meB, in hippocampal DG neuronal engrams following salient contextual fear conditioning.
8. The dynamic expression patterns of Ptprf meA were uncovered, emphasizing the importance of

examining the synaptic function of LAR in relation to PTP δ and PTP σ .

The work is of paramount significance to the field and associated domains. It solidly backs the proposed conclusions and claims. The adopted methodology is sound and robust. The study fulfills the anticipated standards in the field and provides sufficient detail in its methods, ensuring the possibility of replication by other researchers.

One constructive suggestion for this paper is the addition of a comprehensive table, either within the main content or as supplementary material, that encapsulates information regarding the patterns of insertion or absence of meA and meB across different LAR-PTPR family genes, brain regions, cell types, and neural projections. Although a partial summary table is provided in the supplementary section, a more detailed version would greatly enhance the reader's understanding.

Authors' rebuttal letter for Han and Yoon et al., "Specification of neural circuit architecture shaped by context-dependent patterned LAR-RPTP microexons", and description of changes made to the revised manuscript

We appreciate the reviewers' time and effort in evaluating our manuscript. Their detailed comments identified weaknesses in our previous arguments and insufficient explanations of our experiments, and revealed the need for additional evidence. To respond as thoroughly as possible to the reviewers' criticisms, we conducted a series of additional experiments and made relevant changes to the manuscript. We expanded the supplemental data section to accommodate our new findings (there are now 9 main figures and 9 supplementary figures) and repositioned the supplemental figures to maintain their logical relationship with the text and main figures. Below, please find the reviewers' remarks shown in *black italic* typeface and our responses and descriptions of textual modifications in **bold blue** typeface.

REVIEWER COMMENTS

Reviewer #1: *This study from the Ko lab focuses on profiling the spatial regulation of LAR-RPTP microexons in the mouse brain. Targeted RNA-Sequencing of various dissected brain regions as well as RT-PCR validations allowed the quantifications of microexon-A (meA) and microexon-B (meB) across the various LAR-RPTP paralogs, an analysis that didn't result in the observation of striking differences in the inclusion levels of individual microexons across the examined brain regions. The authors further complement this profiling by applying proteomics in an attempt to quantify microexon peptide levels. To investigate if there is cell (sub)type specific LAR-RPTP microexon regulation the authors applied genetic models to sort specific cell types and profiled meA and meB inclusion levels across various brain regions. More interestingly the authors show that Ptpd and Ptpf meA inclusion is promoted by experience-triggered neuronal signaling. The final part of the manuscript takes advantage of Ptpd gene and meA conditional knock-out mouse models that have been recently developed by the Eunjoon Kim laboratory and describes detailed electrophysiological recordings as well as behavioral tests that reveal deficits in object-location memory. Although this study provides some interesting new insights of the in vivo role of Ptpd meA it is rather descriptive and lacks follow-up mechanistic investigation. For example, the authors show an increase of Ptpd meA inclusion in fear-memory activated hippocampal neurons as well as splicing differences in neuronal sub-types, but they fail to address which are the (splicing) factors responsible for this regulation, how do they respond to activity or how their expression changes in different cell types. Furthermore, the novel information provided could be considered as incremental given the previous work from the Kim lab (Park et al., EMBO, 2020). Although this is ultimately an editorial decision, this reviewer thinks that the study would be more suitable for a specialized journal.*

First of all, we respectfully disagree with the reviewer's assessment that the new findings of the current study are merely incremental, particularly when compared to the observations reported by Park et al. These authors did not provide any evidence on the circuit-specific role of PTP δ and its meA microexon in specifying specific synaptic properties and directing a specific behavior (i.e., object-location memory). Park et al. used

PTP δ -knockout mice in which PTP δ was constitutively deleted in most cell-types (e.g., neurons, astrocytes, etc.) residing in both pre- and postsynaptic compartments, and performed field EPSP recordings that identified a ~20% decrease in the input/output (I/O) ratio in the hippocampal SLM but not other hippocampal sublayers (see Figure 2 of Park et al.). We argue that field EPSP measurements could encompass responses from all neurons of the indicated brain region, and do not precisely target those constituting a defined neural circuit. Thus, it is plausible that the reduced I/O ratio documented by Park et al. in the SLM might also reflect an increase in GABAergic synapses and/or tonic GABA release from astrocytes. Overall, these authors cannot claim that their study addressed the circuit-specific role of PTP δ in mice. In contrast, our current study combined circuit-specific PTP δ deletions with electrophysiological and/or behavioral analyses: we deleted PTP δ specifically in subpopulations of hippocampal CA1-projecting neurons in CA3, EC, or SuB and thereby dissected the ‘presynaptic’ role of PTP δ by specifying the specific synaptic properties with patch-clamp configurations. We believe that, the sophisticated circuit-specific manipulations of PTP δ employed in the current study led to distinct electrophysiological phenotypes in the EC-CA1 projections.

Moreover, during the revision, we obtained evidence showing that PTP δ expressed in SuB neurons is required for proper maintenance of NMDA receptor-mediated responses in the CA1 neurons that might mediate object-location memory (presented in new Figure 9 of the revised manuscript). In contrast, Park et al. showed that PTP δ expressed in forebrain regions is responsible for non-REM sleep. We agree that Park et al. demonstrated the physiological significance of the PTP δ meA microexon *in vivo* but believe that this does not diminish the merits of the current study. We do not intend to disparage the findings of Park et al.; rather, based on the abovementioned distinctions, we strongly disagree with the reviewer’s subjective assessment that the findings of the current study are merely incremental. That said, and to address this reviewer’s specific comment #1 (below), we added a sentence to highlight the message of Park et al. in the Introduction section of the revised manuscript. Concerning the identification and validation of the factor(s) that mediate the inclusion/exclusion of PTP δ meA, we take the reviewer’s point and, indeed had mentioned this topic in the Discussion

section of the original manuscript (fourth paragraph) as a natural follow-up that warranted future study. However, we estimate that it will take at least > 3 years to screen candidate regulators for the microexon inclusion/exclusion of LAR-RPTPs *in vitro* (preliminary screening results are provided here only for reviewers’ information) and further validate the results *in vivo*.

We would then need additional time to delineate their significance in region-, cell type-, and circuit-specific contexts. Indeed, such work has been reported for neurexins, spanning 10 years and arising from numerous laboratories. We would like to note that the current study documents the profiling of two major LAR-RPTP microexons across diverse brain regions, cell types, and projections, and thus supports for the first time the physiological significance of the PTP δ meA microexon in regulating specific synapse

properties in a circuit-dependent manner. This was graciously appreciated by the other reviewers. We hope that the reviewer concurs with us, accepts our efforts to address their comments during the revision, and is persuaded to endorse the publication of the current study.

Specific (minor) points:

1. The introduction could be improved by referring to the recent findings regarding the *in vivo* role of *Ptprd meA* (Park et al.,) as well as to previous studies characterizing the molecular function of these microexons (e.g. Yoshida et al., 2012).

This is an excellent suggestion. To address this point, we incorporated the findings of Park et al. in the Introduction section of the revised manuscript. However, we did not further describe the finding of Yoshida et al. in this section because this information was already included in the Discussion section of the original manuscript. This decision was made because, unlike the work of Yoshida et al., the current study did not focus on the molecular interactions between PTP δ meA⁺ variants and postsynaptic ligands. We hope that the reviewer concurs.

2. The authors should provide both kallisto and MAJIQ data for all exons analyzed instead of cherry picking.

We appreciate this suggestion. We initially utilized MAJIQ for the analyses, but noted some disparity between the results obtained using MAJIQ-based analyses and RT-PCR experiments. *Ptprd meA* is composed of two segments, meA1 and meA2. We speculated that this might somehow complicate the determination of exon usages. Thus, we alternatively employed kallisto for analyzing the profiles of *Ptprd meA*. This enabled us to quantify different *Ptprd* variants that lacked meA1, lacked meA2, lacked both, or contained both. We now report on the kallisto analyses for meA and meB of all three LAR-RPTPs in the revised manuscript (presented in Supplementary Figure 2 of the revised manuscript).

3. “*Ptprs meA*⁺ variants were prominently detected in the olfactory bulb..”. This is an overstatement as based on the provided data meA is barely included in *Ptprs*.

We agree with the reviewer’s point and have changed the description as follows in the revised manuscript: “*Ptprs meA*⁺ variants were barely expressed in the examined brain regions, whereas their expression was detectable in the olfactory bulb”.

4. The RT-PCR data provided in Figure 1f, in contrast to their quantification and RNA-Seq data, indicate highest inclusion of *Ptprs meB* in striatum (~50%). I assume this is not a representative picture.

We appreciate the reviewer's careful assessment. In the revised manuscript, we have replaced the indicated RT-PCR result with one that is well aligned with the quantification results.

5. A right bracket should be included

after *Drd2-Cre* (line 7, page 9).

We have done so in the revised manuscript.

6. The structure of the last 2 figures doesn't agree with the text. The text is split into electrophysiological recording and behavioral tests while the figures are split into *Ptprd* gene or microexon KO mouse models.

We appreciate this comment. We have reconfigured the data presented in Figures 6–8 to address the reviewer's constructive suggestions.

7. Cell adhesion molecules should be spelt out in the first line of the discussion.

As instructed, we now spelled out "cell adhesion molecules" in the first line of the Discussion section in the revised manuscript.

Reviewer #2: Han et al. developed a novel method to precisely detect *meA* and *meB* expression and found cell-type specific patterns of *meA* and *meB* in *Ptprd* mRNAs. Alternative splicing of *Ptprd* occurred in an activity-dependent manner. Moreover, conditional depletion of *Ptprd* or *Ptprd meA* in the *SuB/CA1* circuit strikingly increased excitatory synaptic transmission through NMDAR and further decreased objection-location memory. As the biological significance of alternative splicing in neural circuit has been extensively studied and proved for *Nrxn*, the current study does not necessarily present a novel concept. Nevertheless, the authors for the first time addressed and demonstrated the significance of microexon segments, particularly *meA* and *meB* of the LAR-RPTP family, in neural circuit specification.

We greatly appreciate the reviewer's overall positive assessment of our manuscript and enthusiasm for our study, particularly mentioning its significance in the relevant neuroscience field. We hope that the additional data generated during the revision process satisfactorily address the reviewer's remaining concerns.

Major points

1. The authors claimed that LAR-RPTPs utilize microexons to shape combinatorial synaptic adhesion pathways in distinct neural circuits. Indeed, they showed alterations of NMDAR-mediated synaptic transmission and objection-location memory in conditional depletion of *Ptprd* meA in the SuB/CA1 circuit. However, it remains elusive whether the neural connection between SuB and CA1 was altered.

a Experimental strategy to measure connectivity between SuB and hippocampal CA1

We appreciate the reviewer's critical comment. To address this point, we injected WGA-Flpo-AAV and AAV_{2/9}-hSyn-Cre (or AAV_{2/9}-hSyn-ΔCre as a control) into the CA1 and SuB, respectively, of *Ptprd*-meA mutant mice to induce SuB → CA1 circuit-specific *Ptprd* meA deletion, injected AAV₂-retro-mCherry into CA1 at 1 week later, and monitored alteration of the monosynaptic retrograde labeling signaling from CA1 to SuB. We found that deletion of *Ptprd* meA⁺ variants in the SuB → CA1 circuit did not alter its connections of these brain regions, indicating that PTPδ expressed in presynaptic SuB neurons might employ meA to dictate the NMDAR-mediated synaptic transmission in postsynaptic CA1 neurons without compromising its anatomical connectivities. These new data are presented in Supplementary Figure 9 of the revised manuscript.

2. The authors showed an increased *Ptprd* meA inclusion in fear memory engrams of the dentate gyrus (Fig. 5). This activity-dependent alternative splicing was not linked to other findings of synaptic activity or behavior altered by *Ptprd* meA depletion (Fig. 6, 7). In other words, findings shown in Fig. 5 and Fig. 6,7 are independent stories, although each finding is important. If the activity-dependent alternative splicing and the microexon-dependent synaptic activity and behavior are linked, it would strengthen the study.

First, we appreciate the reviewer's assessment that our findings are important. We agree that data presented in Figures 5–7 are seemingly separable. To address the reviewer's insight, we employed the recently developed DREADD agonist, DCZ, in chemogenetics analyses using *Ptprd* meA conditional KI mice. We found that chemogenetic manipulation of subicular neuron excitability could bidirectionally alter *Ptprd* meA profiles, NMDAR-mediated responses, and object-location memory (now presented in Figure 9 of the revised manuscript). These results prompted us to propose a model wherein at SuB→dCA1 circuit PTPδ meA code instructs postsynaptic NMDA receptor-mediated responses and mediates object-location memory (now presented in Figure 9 of the revised manuscript). We hope that the reviewer is satisfied with our new results and concurs with our interpretation.

Minor points

1. Genotype changes of *Ptpred* in SuB induced by WGA-Flopo and AAV-fDIO-Cre should be confirmed in Fig. 6 and 7.

To address the reviewer's point, we injected the indicated viruses into SuB or CA1, collected the infected SuB neurons by FACS, obtained lysates, and performed DNA-PAGE analyses to verify that we had obtained the correct and expected changes in *Ptpred* variants. These new data are included in Figures 6b, 6c and 7b of the revised manuscript.

Reviewer #3: This research utilized intensive transcriptome and proteomics analyses to investigate the insertion patterns of the microexons, meA and meB, in LAR-RTP family proteins. This was done across different brain regions, cell types, and neural projections. The study further delved into the correlation between these patterns and their effects on neural circuit functionality and *in vivo* behavior.

We greatly appreciate the reviewer's overall positive assessment of our paper and hope that our extensive revisions will assuage any remaining concerns.

1. The researchers employed targeted RNA-seq with high sequencing read depth to decipher the

spatiotemporal expression profiles and differential splicing of two microexons — meA and meB — of LAR-RPTP mRNAs.

2. The results from targeted RNA deep sequencing were authenticated using techniques like polyacrylamide gel electrophoresis, quantitative PCR, and Sanger sequencing.

3. PRM mass spectrometry quantification was used to examine PTP δ splice variants containing or lacking meA. It was observed that the abundance of each PTP δ mRNA splice isoform aligns with the corresponding PTP δ proteoforms in three distinct brain areas.

4. The investigation revealed that the meA/meB microexons of LAR-RPTPs exhibit varied expression patterns, even when observed within identical cell types in different brain areas. It was noted that neurons with common long-range projection targets or specific cell connectivity do not consistently use the same microexon codes.

5. The study further showcased that the functions of PTP δ in hippocampal dCA1 circuits are influenced by the presence of an insert at meA. This presence significantly impacts electrophysiological and behavioral attributes in specific mouse models.

6. PTP δ is essential for NMDAR-mediated postsynaptic responses and asynchronous release in a subset of dCA1 circuits.

7. Changes were observed in the expression of Ptp δ meA, but not Ptp δ meB, in hippocampal DG neuronal engrams following salient contextual fear conditioning.

8. The dynamic expression patterns of Ptp δ meA were uncovered, emphasizing the importance of examining the synaptic function of LAR in relation to PTP δ and PTP σ .

We greatly appreciate the reviewer's concise summary of the major findings of our study.

The work is of paramount significance to the field and associated domains. It solidly backs the proposed conclusions and claims. The adopted methodology is sound and robust. The study fulfills the anticipated standards in the field and provides sufficient detail in its methods, ensuring the possibility of replication by other researchers.

We again appreciate the reviewer's enthusiastic support of the publication of our study.

One constructive suggestion for this paper is the addition of a comprehensive table, either within the main content or as supplementary material, that encapsulates information regarding the patterns of insertion or absence of meA and meB across different LAR-PTPR family genes, brain regions, cell types, and neural projections. Although a partial summary table is provided in the supplementary section, a more detailed version would greatly enhance the reader's understanding.

As the reviewer suggested, we added a new Supplementary Table (Supplementary Table 5 in the revised manuscript) encapsulating information on the profiling of LAR-RPTP microexons across the brain regions, neuron types, and neural circuits analyzed in the current study.

Again, we thank the reviewers for their careful assessment of our paper and hope that the revised manuscript is now acceptable for publication in *Nature Communications*.

REVIEWERS' COMMENTS

Reviewer #1 (Remarks to the Author):

The authors have effectively addressed my comments and I thus endorse publication of their study in Nature Communications.

Reviewer #2 (Remarks to the Author):

The authors adequately addressed my concerns. Particularly their findings responding to my concern #2 provided a biological link between the activity-dependent alternative splicing and the microexon-dependent synaptic activity and behavior.

Reviewer #3 (Remarks to the Author):

The authors complied with my suggestion and included Supplementary Table 5, which fully satisfies my request.

Authors' rebuttal letter for Han and Yoon et al., "Specification of neural circuit architecture shaped by context-dependent patterned LAR-RPTP microexons", and description of changes made to the revised manuscript

We appreciate the reviewers' time and effort in evaluating our revised manuscript. Below, please find the reviewers' remarks shown in *black italic* typeface and our responses in **bold blue** typeface.

REVIEWER COMMENTS

Reviewer #1: *The authors have effectively addressed my comments and I thus endorse publication of their study in Nature Communications.*

We greatly appreciate the support of the reviewer for publication.

Reviewer #2: *The authors adequately addressed my concerns. Particularly their findings responding to my concern #2 provided a biological link between the activity-dependent alternative splicing and the microexon-dependent synaptic activity and behavior.*

We greatly appreciate the support of the reviewer for publication.

Reviewer #3: *The authors complied with my suggestion and included Supplementary Table 5, which fully satisfies my request.*

We greatly appreciate the support of the reviewer for publication.